EMBO
Molecular Medicine

# *Nanog* maintains stemness of *Lkb1*-deficient lung adenocarcinoma and prevents gastric differentiation

Xinyuan Tong[1], Yueqing Chen[1,2], Xinsheng Zhu[3], Yi Ye[4], Yun Xue[1,2], Rui Wang[5,6], Yijun Gao[1], Wenjing Zhang[1], Weiqiang Gao[7,8], Lei Xiao[9], Haiquan Chen[5,6], Peng Zhang[3,*] & Hongbin Ji[1,2,3,4]

## Abstract

Growing evidence supports that *LKB1*-deficient *KRAS*-driven lung tumors represent a unique therapeutic challenge, displaying strong cancer plasticity that promotes lineage conversion and drug resistance. Here we find that murine lung tumors from the *Kras*[LSL-G12D/+]; *Lkb1*[flox/flox] (KL) model show strong plasticity, which associates with up-regulation of stem cell pluripotency genes such as *Nanog*. Deletion of *Nanog* in KL model initiates a gastric differentiation program and promotes mucinous lung tumor growth. We find that NANOG is not expressed at a meaningful level in human lung adenocarcinoma (ADC), as well as in human lung invasive mucinous adenocarcinoma (IMA). Gastric differentiation involves activation of Notch signaling, and perturbation of Notch pathway by the γ-secretase inhibitor LY-411575 remarkably impairs mucinous tumor formation. In contrast to non-mucinous tumors, mucinous tumors are resistant to phenformin treatment. Such therapeutic resistance could be overcome through combined treatments with LY-411575 and phenformin. Overall, we uncover a previously unappreciated plasticity of *LKB1*-deficient tumors and identify the Nanog-Notch axis in regulating gastric differentiation, which holds important therapeutic implication for the treatment of mucinous lung cancer.

**Keywords** drug resistance; gastric differentiation; LKB1; *Nanog*; Notch

**Subject Categories** Cancer; Respiratory System; Stem Cells & Regenerative Medicine

## Introduction

Tissue-specific lineage trans-differentiation events such as gastric differentiation have been recently observed in human non-small cell lung cancer (NSCLC). The transcription factor NKX2-1 is generally expressed in human lung adenocarcinoma (ADC), but its expression decreases with poor differentiation and malignant progression (Winslow *et al*, 2011; Li *et al*, 2014; Cha & Shim, 2017; Boland *et al*, 2018). Previous studies have shown that loss of *Nkx2-1* in *Kras*-driven lung cancer genetically engineered mouse model (GEMM) triggered the loss of pulmonary identity and promotes an unexpected gastric differentiation (Maeda *et al*, 2012; Snyder *et al*, 2013). Interestingly, these tumors mimic human lung invasive mucinous adenocarcinoma (IMA) expressing gastric markers such as *Hnf4α*, a master regulator of gastrointestinal differentiation (Snyder *et al*, 2013; Camolotto *et al*, 2018). Chromatin immunoprecipitation sequencing (ChIP-seq) analysis demonstrates that NKX2-1 transcriptionally regulates MUC5AC (mucin 5AC, oligomeric mucus/gel-forming) expression by binding to its promoter (Maeda *et al*, 2012). Meanwhile, HNF4α is suppressed by NKX2-1-mediated tissue-specific FOXA1/FOXA2 engagement (Minoo *et al*, 2007; Gao *et al*, 2008; Wederell *et al*, 2008; Snyder *et al*, 2013; Camolotto *et al*, 2018). In the context of *NKX2-1* deficiency along with *KRAS*[G12D] mutation, SPDEF and FOXA3 promote mucin genes expression and recapitulate human IMA histopathology (Park *et al*, 2007; Guo *et al*, 2017). IMA is genetically and therapeutically distinct from other lung ADC subtypes (Finberg *et al*, 2007; Li *et al*, 2014; Nakaoku *et al*, 2014; Shim *et al*, 2015; Luo *et al*, 2016; Cha & Shim, 2017; Boland *et al*, 2018). IMA frequently harbor "undruggable" *KRAS* mutations but rarely *EGFR* mutations and are unlikely responsive to tyrosine kinase inhibitors (TKIs) (Finberg *et al*, 2007; Boland *et al*, 2018).

1   State Key Laboratory of Cell Biology, Shanghai Institute of Biochemistry and Cell Biology, Center for Excellence in Molecular Cell Science, Chinese Academy of Sciences, Shanghai, China
2   University of Chinese Academy of Sciences, Beijing, China
3   Department of Thoracic Surgery, Shanghai Pulmonary Hospital, Tongji University School of Medicine, Shanghai, China
4   School of Life Science and Technology, Shanghai Tech University, Shanghai, China
5   Department of Thoracic Surgery, Fudan University Shanghai Cancer Center, Shanghai, China
6   Department of Oncology, Shanghai Medical College, Fudan University, Shanghai, China
7   State Key Laboratory of Oncogenes and Related Genes, Shanghai Cancer Institute, Renji Hospital, Shanghai Jiao Tong University School of Medicine, Shanghai, China
8   Med-X Research Institute, Shanghai Jiao Tong University, Shanghai, China
9   College of Animal Science and Zhejiang University School of Medicine, Zhejiang University, Hangzhou, China
    *Corresponding author. Tel: +86 21 65115006; E-mail: zhangpeng1121@tongji.edu.cn
    **Corresponding author. Tel: +86 21 54921108; E-mail: hbji@sibcb.ac.cn

Although recent efforts have established several mouse models for mucinous tumors (Fisher *et al*, 2001; Maeda *et al*, 2012; Lee *et al*, 2013; Schuster *et al*, 2014; Skoulidis *et al*, 2015; Serresi *et al*, 2016; Guo *et al*, 2017), the pathogenesis of IMA and potential therapeutics still await further investigation.

*LKB1/STK11*, encoding a serine/threonine kinase implicated in energy homeostasis, is one of the leading mutated genes in NSCLC (Sanchez-Cespedes *et al*, 2002; Shaw *et al*, 2004; Mahoney *et al*, 2009; Gao *et al*, 2010; Fang *et al*, 2014; Skoulidis *et al*, 2015). Accumulating evidences support that concurrent genetic alteration of *KRAS* and *LKB1* defines a unique molecular subtype with potent cancer plasticity, exhibiting high metastatic competence, frequent therapeutic resistance and poor clinical prognosis, and thus represents a major challenge for lung cancer therapeutics (Mahoney *et al*, 2009; Skoulidis *et al*, 2015). This oncological genotype imposes a metabolic vulnerability related to the dependence on pyrimidine metabolism (Kim *et al*, 2017). GEMM studies highlight the strong plasticity of $Kras^{LSL-G12D/+}$; $Lkb1^{flox/flox}$ (KL) lung tumors through lineage transition from adenocarcinomas (ADC) to squamous cell carcinomas (SCC), and its link with therapeutic resistance (Ji *et al*, 2007; Li *et al*, 2015). Although the mechanism remains unclear, we reason that LKB1 deficiency may confer strong stemness to the lung cancer cells, allowing trans-differentiation from ADC to SCC. We have previously found that excessive accumulation of oxidative stress induced by *Lkb1* deletion and metabolic reprogramming during tumor progression plays important roles in this phenotypic transition. Moreover, the Hippo pathway and epigenetic factors including EZH2 also contribute to the squamous trans-differentiation (Gao *et al*, 2014; Li *et al*, 2015; Huang *et al*, 2017; Zhang *et al*, 2017). Nonetheless, the link between stemness of KL tumors and strong plasticity remains largely elusive. The roles of LKB1 in regulating self-renewal of skeletal muscle progenitor cells (Shan *et al*, 2014; Shan *et al*, 2017) and hematopoietic stem cell survival have been reported (Gan *et al*, 2010; Gurumurthy *et al*, 2010; Nakada *et al*, 2010). Investigating the molecular determinants governing the *LKB1*-deficient cancer stemness is of particular importance for better understanding of KL tumor plasticity as well as providing novel therapeutic strategy.

NANOG is an important transcription factor maintaining the regulatory network responsible for embryonic stem cell self-renewal and pluripotency (Cavaleri & Scholer, 2003; Mitsui *et al*, 2003; Hyslop *et al*, 2005; Silva *et al*, 2009). Also, NANOG acts as a cancer stemness marker and promotes cancer tumorigenesis and stemness (Noh *et al*, 2012a; Noh *et al*, 2012b; Chen *et al*, 2016; Lu *et al*, 2018; Zhang *et al*, 2018). Aberrant NANOG expression is commonly found in multiple cancer types including lung ADC (Watanabe, 2009; Du *et al*, 2013; Vaira *et al*, 2013; Li *et al*, 2013a; Li *et al*, 2013b; Liu *et al*, 2014; Park *et al*, 2016; Zhao *et al*, 2018). Moreover, NANOG is known for critically contributing to tumor initiation and epithelial–mesenchymal transition (Chiou *et al*, 2010; Yin *et al*, 2015). High NANOG expression is associated with poor tumor differentiation and advanced tumor stage (Du *et al*, 2013; Li *et al*, 2013a; Li *et al*, 2013b; Liu *et al*, 2014; Park *et al*, 2016). A recent study also demonstrates that NANOG elicits a lineage-restricted mitogenic function in squamous cell carcinomas from stratified epithelia (Piazzolla *et al*, 2014). Understanding the roles of NANOG in cancer stemness and lineage switch may benefit therapeutic development for a wide range of human diseases.

We find here that *Lkb1*-deficient lung tumors from the KL model harbor stronger stemness and display significant up-regulation of Nanog expression, in contrast to $Kras^{G12D/+}$ or $Kras^{G12D/+}$; $P53^{-/-}$ tumors. Genetic ablation of *Nanog* in the KL model triggers gastric differentiation and produces mucinous lung tumors through Notch signaling activation. Such lineage transition renders lung tumors resistant to phenformin treatment, which could be overcome via combinational treatments with γ-secretase inhibitor LY-411575.

# Results

## Nanog as an important factor in maintaining KL tumor stemness

To investigate the stemness property in relation to LKB1 deficiency, we comparatively analyzed the RNA-Seq data from lung ADC from $Kras^{LSL-G12D/+}$ (K), $Kras^{LSL-G12D/+}$; $Lkb1^{flox/flox}$ (KL), and $Kras^{LSL-G12D/+}$; $P53^{flox/flox}$ (KP) mouse models, in which the oncogenic KrasG12D allele is conditionally activated following Ad-Cre infection by nasal inhalation as described previously (Li *et al*, 2015). We found that stem cell pluripotency expression pattern was significantly enriched in KL tumors in contrast to K or KP tumors (Fig 1A and B). Notably, *Nanog*, the well-established factor maintaining stem cell property (Mitsui *et al*, 2003), was remarkably increased in KL tumors (Fig 1C). We further confirmed this observation through IHC analysis (Fig 1D).

We hypothesized that *Nanog* might be a crucial factor for maintaining KL tumor stemness. We thus generated *Nanog* conditional knockout mice (Fig EV1) and crossed them with various *Kras* models to obtain $Kras^{LSL-G12D/+}$; $Nanog^{flox/flox}$ (KN), $Kras^{LSL-G12D/+}$; $P53^{flox/flox}$; $Nanog^{flox/flox}$ (KPN), and $Kras^{LSL-G12D/+}$; $Lkb1^{flox/flox}$; $Nanog^{flox/flox}$ (KLN) cohorts (Fig 1E). We treated the mice with Ad-Cre and analyzed lung tumor incidence. There were no significant alterations on either the tumor burden or tumor number following *Nanog* deletion in K, KL, or KP mice compared with control mice (Fig EV2A–C). Detailed pathological analyses revealed no impact of *Nanog* deletion in K or KP lung tumors, both of which displayed classical ADC pathology (Fig 1F and G). Interestingly, the deletion of *Nanog* in KL model promoted the formation of mucinous lung tumors (Fig 1H). In KLN mice, we found more small adenomas but there was no significant change in ADC tumorigenesis compared with KL mice (Fig EV2D and E). Meanwhile, there was no impact on squamous differentiation (Ji *et al*, 2007; Li *et al*, 2015) (Fig EV2F). Considering the specific up-regulation of *Nanog* in KL tumors, these observations suggest that *Nanog* might serve as an important factor for maintaining cancer stemness triggered by LKB1 deficiency.

## Nanog deletion in KL tumors triggers gastric differentiation

IHC staining and qPCR analysis verified the efficient knockout of *Nanog* in KL mice (Fig EV3A and B). We then pathologically and molecularly characterized the mucinous lung tumors from KLN mice. Human IMA tumor cells contained abundant intracytoplasmic mucin admixed with invasive adenocarcinoma patterns (Fig 2A). Similarly, KLN mucinous tumors were composed of columnar or goblet cell morphology with mucus in apical cytoplasm and showed small basal oriented nuclei (Fig 2B). We observed that these tumors showed the same heterogeneous mixture of acinar or papillary growth patterns as in non-mucinous

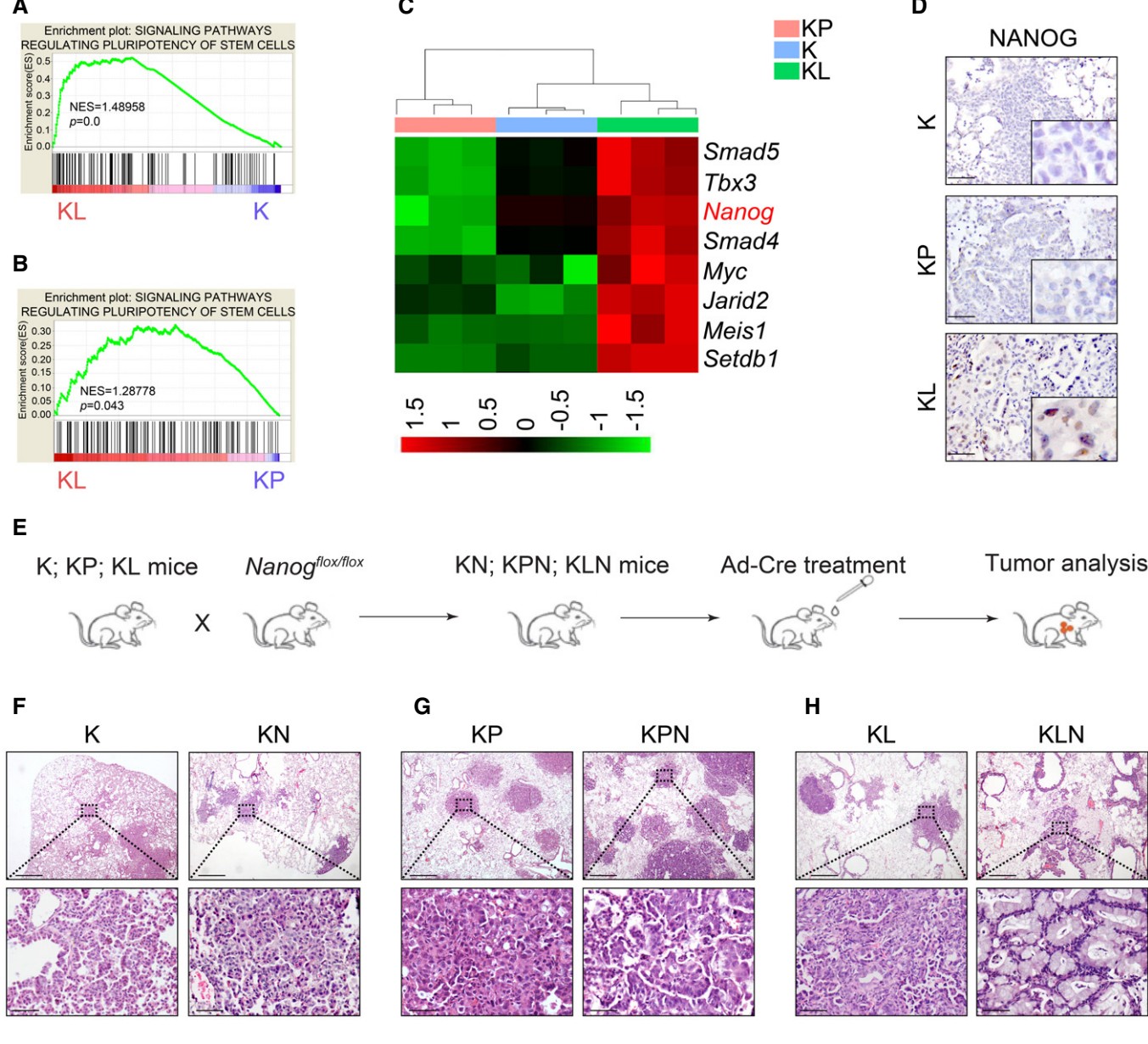

**Figure 1. Identification of *Nanog* as the stemness factor in KL lung tumors.**

A, B  Gene Set Enrichment Analysis (GSEA) identifies stem cell-like expression signature in KL tumors compared with *Kras*$^{G12D/+}$ (K) lung tumor (A) or *Kras*$^{G12D/+}$; *P53*$^{-/-}$ (KP) lung tumors (B). Significance was calculated by permutation test.

C  Heat map demonstration of genes that were mostly enriched in the pathway of pluripotency of stem cells. The color bar showed the relative RNA-Seq signal (Z-score of normalized FPKM).

D  The protein abundance of NANOG was up-regulated in KL tumors compared with K and KP mice. Scale bar, 50 μm.

E  Scheme illustrator of mouse model.

F–H  Histological examinations of tumor types in (F) K versus *Kras*$^{G12D}$; *Nanog*$^{L/L}$ (KN), (G) KP versus *Kras*$^{G12D}$; *P53*$^{L/L}$; *Nanog*$^{L/L}$ (KPN), and (H) KL versus *Kras*$^{G12D}$; *Lkb1*$^{L/L}$; *Nanog*$^{L/L}$ (KLN). Tumor tissues collected form indicated mice analyzed using hematoxylin and eosin (H&E) staining. Representative H&E images are shown. Scale bar, 500 μm (top), 50 μm (bottom).

tumors. Human and murine IMA were both invasive and lacked a circumscribed border with miliary spread into adjacent lung parenchyma. Consistent with the histological features, the reactivity of tumor tissues with Alcian blue (AB) and Periodic acid–Schiff (PAS) dyes confirmed the presence of acidic polysaccharides, mucopolysaccharides, and neutral mucins, typical of human mucinous carcinomas (Hollingsworth & Swanson, 2004). *Nanog* deletion in KL mice significantly promoted the formation of IMA (Fig 2C). The mucinous differentiation was not detectable at 4 weeks of Ad-Cre treatment but was clearly present in around 15% tumors at 8 weeks (Fig 2D and E). The precursor lesions such as atypical alveolar hyperplasia (AAH) and adenoma (AD)

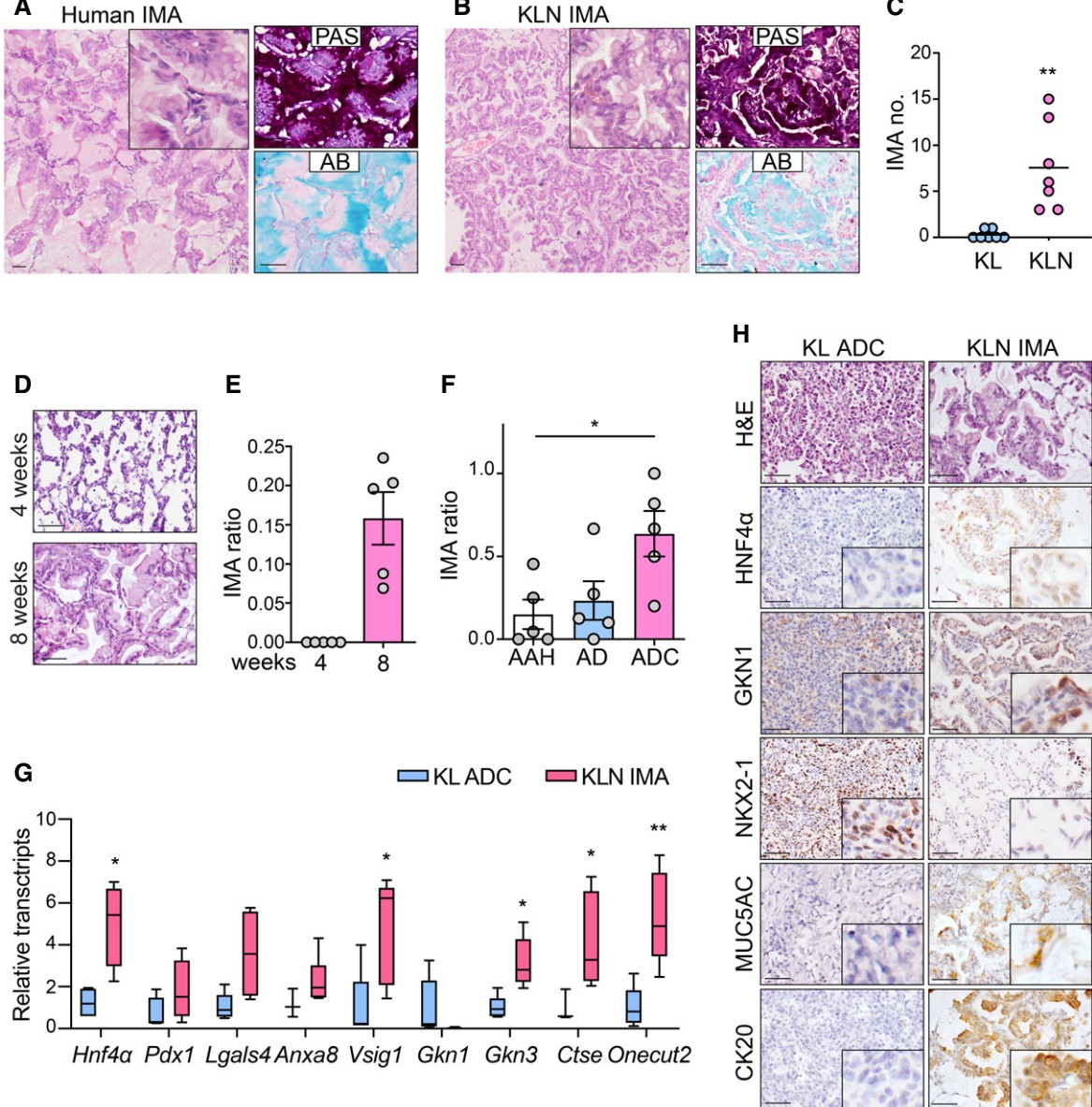

**Figure 2. *Nanog* deletion triggers gastric differentiation in KL model.**

A, B Human (A) and KLN murine (B) lung sections were stained with H&E or with Periodic acid–Schiff (PAS) and Alcian blue (AB). Scale bar, 50 μm.

C Statistical analysis of mucinous tumor numbers (no.) in KL (*n* = 7) or KLN (*n* = 7) mice. Significance was calculated by two-tailed unpaired Student's *t*-test with Welch's correction. Results were shown as mean. **P* = 0.0066.

D Representative pictures of lung tumors at 4 weeks and 8 weeks post-Ad-Cre infection in KLN mice. Scale bar, 50 μm.

E Statistical analysis of IMA ratio at 4 and 8 weeks post-Ad-Cre treatment (*n* = 5 per group). Results were shown as mean ± SEM.

F Statistical analysis of ratio of mucinous phenotype in AAH, AD, and ADC (*n* = 5 per group). Results were shown as mean ± SEM. Significance was calculated by one-way ANOVA with Dunnett's multiple comparisons test. **P* = 0.0215.

G qPCR analysis of stomach-restricted gene set between KL ADC and KLN mucinous tumor sections (*n* = 3 or 4 per group). Results were shown in a box and whisker plot with the minimum value, first quartile, median, third quartile, and maximum value. The box represented the 50% of the central data, with a line inside represented the median. The significance was calculated by two-tailed unpaired Student's *t*-test with Welch's correction. **P* = 0.0260 (*Hnf4α*), **P* = 0.0275 (*Vsig1*), **P* = 0.0122 (*Gkn3*), **P* = 0.0324 (*Ctse*), ***P* = 0.0084 (*Onecut2*).

H Representative histological and IHC staining for HNF4α, GKN1, NKX2-1, MUC5AC, CK20 in lung tumors derived from ADC and IMA. Scale bar, 50 μm.

were occasionally present with mucin, whereas most ADC were evidently mucinous (Fig 2F). Non-mucinous ADC in KLN mice expressed type II pneumocyte marker pro-surfactant protein C (proSPC) or Club cell 10-kDa protein (CC10), whereas mucinous tumors only expressed CC10 (Fig EV3C). TP63, the basal cell marker for SCC, was also absent in mucinous ADC (Fig EV3C).

These data indicate that Club cells might serve as the cellular origin for mucinous ADC.

We next determined whether KLN tumors adopted the gastric cell fate conversion pattern as previously described (Maeda *et al*, 2012; Snyder *et al*, 2013; Camolotto *et al*, 2018). We evaluated a gene set correlated with gastric differentiation, including *Hnf4α, Pdx1, Lgal-s4, Anxa8, Vsig1, Gkn1, Gkn3, Ctse,* and *Onecut2* (Snyder *et al*, 2013). We found that *Hnf4α, Vsig1, Gkn3, Ctse,* and *Onecut2* were

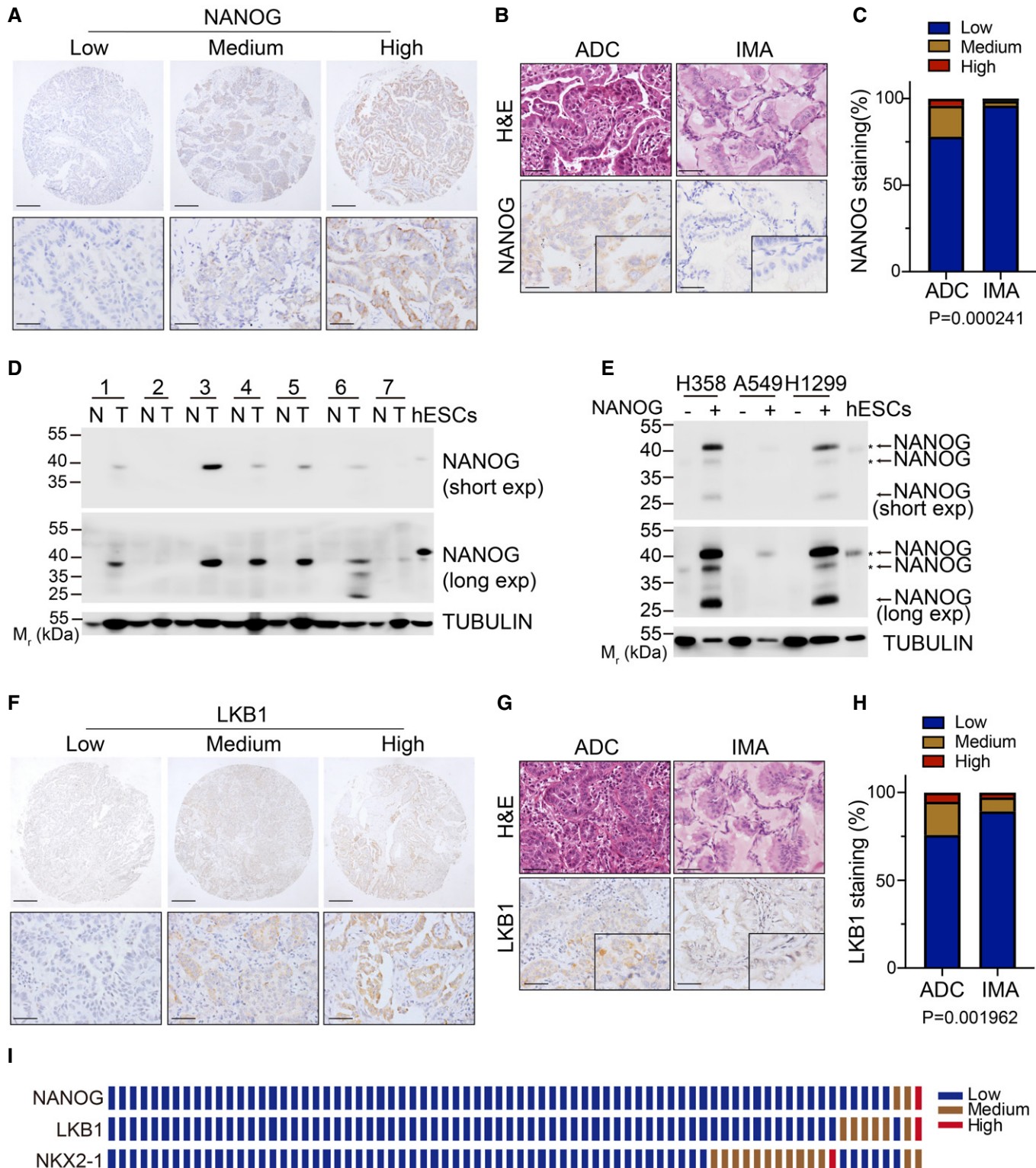

Figure 3.

◀

**Figure 3. Frequently concurrent loss of NANOG, LKB1, and NKX2-1 expression in human IMA.**

A  Representative images of low, medium, high protein expression of NANOG in human lung adenocarcinomas. Scale bar, 150 μm (top), 50 μm (bottom).
B  Representative H&E and IHC staining for NANOG in human ADC and IMA. Scale bar, 50 μm.
C  Statistic analysis of NANOG expression in ADC and IMA. The tumor tissues were analyzed by IHC staining and graded as low, medium, high expression. Significance was calculated by $\chi 2$ test for trend.
D  Western blot for NANOG and TUBULIN in 7 paired human normal lungs (N) and ADC samples (T) and human ESCs. $M_r$, relative molecular mass. kDa, killoDalton.
E  Western blot for NANOG and TUBULIN in human adenocarcinoma cell lines transfected with empty vector or human NANOG overexpression vector. The arrow indicated the band of exogenous human NANOG. The asterisk indicated the band of endogenous human NANOG. $M_r$, relative molecular mass. kDa, killoDalton.
F  Representative images of low, medium, high protein expression of LKB1 in human lung adenocarcinomas. Scale bar, 150 μm (top), 50 μm (bottom).
G  Representative H&E and IHC staining for LKB1 in human ADC and IMA. Scale bar, 50 μm.
H  Statistic analysis of LKB1 expression in ADC and IMA. The tumor tissues were analyzed by IHC staining and graded as low, medium, high expression. Significance was calculated by $\chi^2$ test for trend.
I  The expression grade of IHC staining of NANOG, NKX2-1, LKB1 in 76 human IMA (1 sample/row).

Source data are available online for this figure.

significantly up-regulated in mucinous KLN tumors (Fig 2G). We used IHC to evaluate the expression pattern of gastric and pulmonary proteins in mucinous and non-mucinous tumors (Fig 2H). Gastric proteins HNF4α and GKN1 were increased in mucinous tumors as previously reported (Snyder *et al*, 2013). On the contrary, NKX2-1 was decreased in mucinous tumors which was consistent with its inhibitory role in gastric differentiation (Maeda *et al*, 2012) (Fig 2H). These lines of evidence imply that *Nanog* deletion in KL tumors induces the expression of stomach-restricted genes and exhibits gastric differentiation. We also examined established biomarkers for mucinous lung tumors, such as CK20 (cytokeratin 20) and MUC5AC, which were specifically expressed with strong intensity in mucinous lung tumors in contrast to no expression in non-mucinous ADC from KL mice (Fig 2H).

### Frequent concurrent loss of NANOG and LKB1 expression in human mucinous lung tumors

To determine the expression of NANOG in human lung tissues, we collected 175 human lung ADC and 76 IMA samples to perform IHC staining. In contrast to the nuclear location detected in the mouse model (Fig 1D), NANOG was mainly expressed in the cytoplasm of human lung cancer cells (Fig 3A), indicative of different expression patterns in different species. Although the majority of NANOG IHC staining were not strong, we classified NANOG expression into three groups with relative low, medium, high level as previously described (Gao *et al*, 2014) (Fig 3A). We next investigated the difference of NANOG expression between human lung ADC and IMA (Fig 3B and C). Statistical analysis showed that 21.7% (38/175) of ADC displayed relative high or medium expression of NANOG, whereas only 1.3% (1/76) of IMA showed high NANOG expression. While 2.6% (2/76) showed medium NANOG level, about 96% (73/76) of the IMA showed low NANOG level (Fig 3C). We observed that NANOG could be detected using Western blot in most of the human lung ADCs with relative medium or high level of IHC staining (Fig 3D). When we overexpressed full-length human NANOG cDNA in human lung cancer cell lines, we found that the exogenous NANOG expression produced 3 different bands, two around 40 kDa and another one around 25 kDa (Fig 3E). Western blot of one cell line H358 exhibited the 40 kDa lower band, which was also observed in human NANOG overexpression groups (Fig 3E). However, the endogenous NANOG band (about 40kDa) detected in human embryonic stem cells (hESCs) was not seen in human lung

ADCs or in cell lines (Fig 3D and E). We reason that such difference might be due to different post-translational modifications of NANOG in different context and/or to the fact that the functional NANOG seen in human ESCs is not expressed in human lung ADC and cell lines. We also checked NANOG transcriptional level in human lung ADC cell lines from public database (EBML-EBI expression atlas) and found that NANOG was expressed at an extremely low level with < 1 TPM/RPKM (Table EV1).

We next investigated the difference of LKB1 expression between ADC and IMA (Fig 3F and G). Statistical analysis showed that 24% (42/175) of ADC displayed expression of LKB1, whereas 9.1% (7/76) of IMA showed LKB1 expression (Fig 3H). We further detected the protein abundance of NKX2-1 and LKB1 in matching IMA tissues. About 90% (69/76) of IMA displayed low LKB1 level and 81% (62/76) of IMA showed low expression of NKX2-1 (Fig 3I). Interestingly, about 98.6% (68/69) of IMA with low *LKB1* expression showed low NANOG expression, indicating the frequent concurrent down-regulation of these two genes. Indeed, simultaneous low expression of NANOG and LKB1 was observed in over 89% (68/76) human lung IMA (Fig 3I). These results together with the findings from mouse models indicate that the frequent loss of NANOG and LKB1 might functionally regulate the development of mucinous lung tumors.

### Perturbation of Notch signaling inhibits gastric differentiation

To gain insight into the underlying mechanism, we examined several well-established pathways committed in cell lineage conversion, including Notch signaling, Wnt pathway, and Hedgehog network (Garcia Campelo *et al*, 2011). In contrast to non-mucinous tumors, the mucinous tumors from KLN mice displayed significant up-regulation of Notch pathway components including *Hes1*, *Hes5*, *Hey1* and *Hey2*, but not Wnt nor Hedgehog signaling (Fig 4A). It has been documented that Notch signaling promoted goblet cells differentiation and mucin production, and inhibited the differentiation of ciliated cells in both developmental and mature lung epithelia (Guseh *et al*, 2009; Tsao *et al*, 2009; Rock *et al*, 2011; Whitsett & Kalinichenko, 2011; Gomi *et al*, 2015; Yao *et al*, 2018). Consistently, the protein abundance of Notch1 intracellular domain (NICD), the major activator of canonical Notch signaling pathway, as well as Notch effector HES1, were markedly increased in KLN mucinous tumors relative to non-mucinous tumors, as evidenced by IHC staining and statistical analyses (Fig 4B–D). Clinically, higher NICD

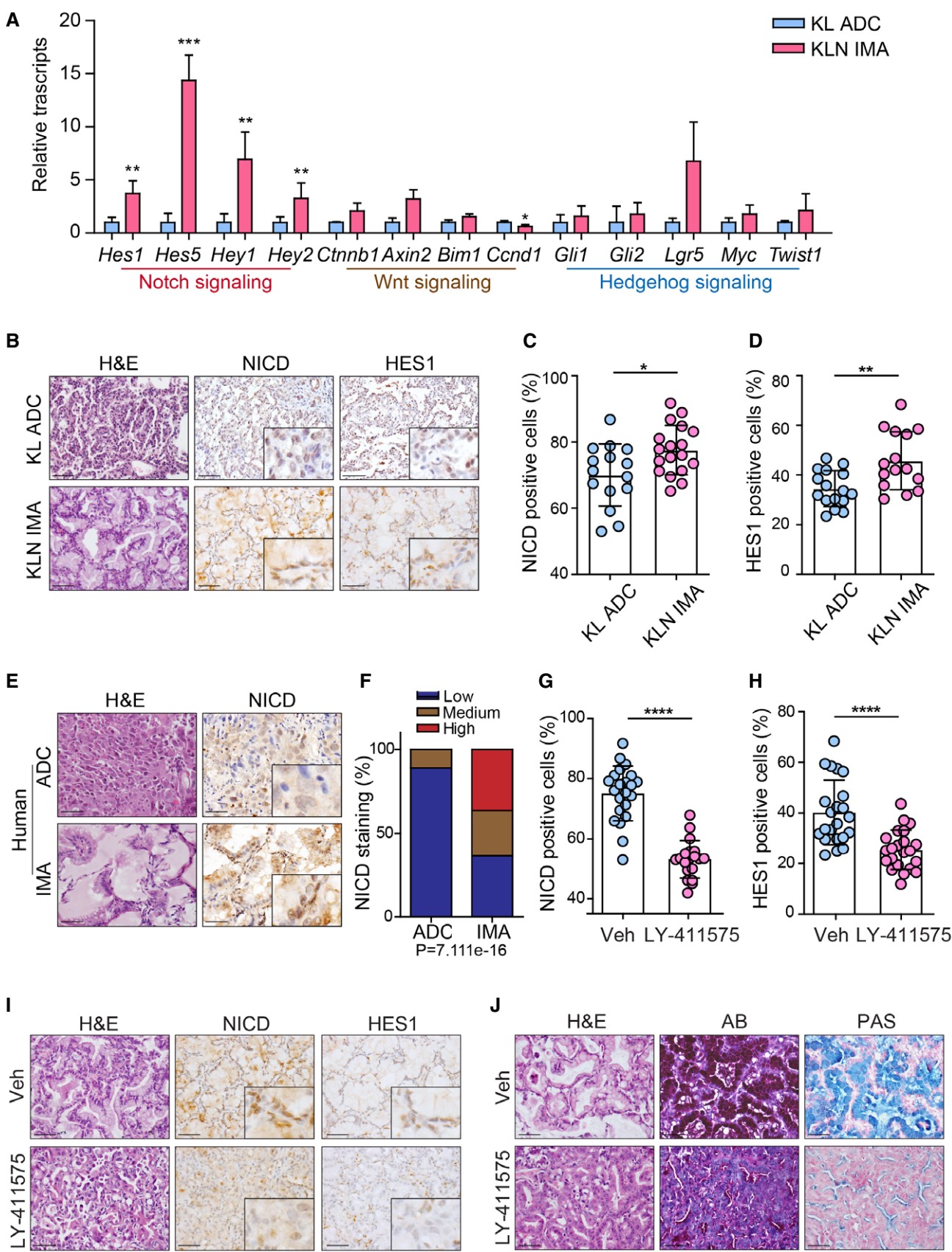

**Figure 4.**

**Figure 4.  Blockage of Notch signaling prevents mucinous differentiation in KLN mice.**

A       Relative transcripts of components of Notch signaling, Wnt signaling, Hedgehog signaling pathway in IMA tumors of KLN group relative to KL mice ($n = 4$ per group). Results are mean $\pm$ SD. Significance was calculated by two-tailed unpaired Student's $t$-test with Welch's correction. **$P = 0.0048$ (*Hes1*), ***$P = 0.0003$ (*Hes5*), **$P = 0.0051$ (*Hey1*), **$P = 0.0025$ (*Hey2*), *$P = 0.0110$ (*Ccnd1*).

B–D    H&E staining of tumors, IHC staining (B) and quantification of NICD (C) and HES1 (D) in KL ADC and KLN IMA. Representative images were shown. Scale bar, 50 $\mu$m. Results were shown as mean $\pm$ SD. The numbers of tumors analyzed over KL ADC and KLN IMA were 15, 18 for NICD, 16, 15 for HES1. Significance was calculated by two-tailed unpaired Student's $t$-test with Welch's correction. *$P = 0.0188$ (C), **$P = 0.0040$ (D).

E       Representative H&E of tissues and IHC staining for NICD in human ADC and IMA. Scale bar, 50 $\mu$m.

F       Statistic analysis of NICD expression in human IMA relative to ADC. The tumor tissues were analyzed by IHC staining and graded according as low, medium, or high expression. Significance was calculated by $\chi 2$ test for trend.

G, H    Statistical analysis of NICD (G) and HES1 (H) expression in lung sections of KLN mice following vehicle or LY-411575 treatment. The numbers of tumors analyzed over vehicle and LY-411575 groups were 22, 19 for NICD, 22, 22 for HES1. Results were shown as mean $\pm$ SD. Significance was calculated by two-tailed unpaired Student's $t$-test with Welch's correction. ****$P < 0.0001$.

I       H&E and IHC staining of NICD and HES1 in lung tumor tissues of vehicle and LY-411575 groups. Scale bar, 50 $\mu$m.

J       H&E, AB and PAS staining in lung tumors of vehicle and LY-411575 groups. Scale bar, 50 $\mu$m.

expressions were observed in human IMAs in contrast to ADCs (Fig 4E and F). We evaluated expression of HES1, HES5, and HEY1 in human lung IMA. We found that only HES5 was expressed in around 50% of human ADC and 87% of human IMA (Fig EV4A). Statistical analysis indicated an increased expression of HES5 in human IMA (Fig EV4B). We further found that HES5 was expressed in 68.7% of IMA with concurrent loss of NANOG, LKB1, and NKX2-1 (Fig EV4C). Collectively, our data indicate that Notch signaling is potentially committed to the mucinous tumor lineage specification.

Proteolytic processing of Notch by $\gamma$-secretase is an essential step for Notch pathway activation (Ntziachristos *et al*, 2014). It has been reported that LY-411575, a commonly used $\gamma$-secretase inhibitor, repressed Notch pathway in tumor cells (Maraver *et al*, 2012; Ambrogio *et al*, 2016). We treated KLN mice with LY-411575 (3 mg/kg daily) via gavage for 2 weeks. Pathological analyses showed that LY-411575 treatment led to significantly decreased expression of NICD and HES1 in lung tumors (Fig 4G–I), indicative of impaired Notch signaling. We found a robust reduction of mucin production in KLN mice following Notch inhibitor therapy compared with control group (Figs 4J and EV5A), despite the absence of significant change in the total tumor number and burden (Fig EV5B and C). These data suggest that LY-411575 therapy mainly prevented mucinous differentiation without sufficient influence on tumor progression. Accordingly, IHC staining of proliferation marker Ki67 and apoptotic marker cleaved caspase-3 (CC3) indicated that LY-411575 treatment did not affect cell proliferation nor apoptosis in ADC or mucinous tumors (Fig EV5D–F).

### Mucinous tumors are resistant to phenformin treatment

Phenformin, an analog of anti-diabetes drug metformin, has been identified to selectively target *LKB1*-deficient ADC (Shackelford *et al*, 2013). To confirm the efficiency of phenformin in KL context, tumor-bearing KL mice were administrated phenformin (1.8 mg/ml) in drinking water for 2 weeks and subjected to pathological examination. We found that phenformin treatment resulted in the decrease of ADC tumor numbers (Fig 5A), blocked tumor cell proliferation, and induced cell apoptosis in KL ADC (Fig 5B–D). We also treated KLN mice with phenformin using the same strategy. The number of ADC was decreased after phenformin treatment (Fig 5E). However, little effect was observed on the occurrence of mucinous tumors after phenformin treatment in KLN mice (Fig 5F and G). We

further checked cell proliferation and apoptosis in tumors (Fig 5H–J). Anti-Ki67 staining revealed that ADC cells displayed an obvious decrease of proliferation rate, whereas the mucinous tumors remained proliferative (Fig 5I). In addition, a higher apoptosis signature evidenced by the increase of CC3-positive cells was observed in non-mucinous ADC, but not in IMA (Fig 5J). Together, these data demonstrate that mucinous differentiation renders *Lkb1*-deficient tumors resistant to phenformin treatment.

### Combined LY-411575 treatment overcomes the phenformin resistance in mucinous tumors

Given the limited response of phenformin in mucinous tumors, we investigated whether simultaneous perturbation of Notch pathway would improve the therapeutic efficacy in *Lkb1*-deficient tumors. KLN mice were given a combinational therapy consisting of $\gamma$-secretase inhibitor LY-411575 and phenformin for 2 weeks following tumor induction. Pathological analysis showed that most tumors were negative for PAS and AB staining in combinational treatment group (Fig 6A). Consistently, the number of IMA was decreased following drug treatment (Fig 6B). Furthermore, we found that combinational therapy notably reduced the tumor size and average tumor numbers (Fig 6C and D). These observations were further supported by the analysis of the proportion of Ki67- or CC3-positive cells in tumor sections (Fig 6E–G). Compared with control group or single-agent therapy, combinational therapy significantly inhibited cell proliferation and induced apoptosis (Fig 6E and F). These results indicate that concomitant treatment of LY-411575 and phenformin provides novel intervention strategy for KLN mucinous tumors, and overcomes the treatment resistance imposed by mucinous lineage transition (Fig 6H).

## Discussion

KL tumors are defined as a unique molecular subtype and are notorious for their strong tumor plasticity, allowing multiple cell lineage trans-differentiation programs including ADC, SCC, and mixed adeno-squamous cell carcinoma (Ad-SCC) (Ji *et al*, 2007). However, the molecular mechanisms underlying the stemness of KL tumors in relation to their phenotypic plasticity still remain largely unknown. In this study, we reveal that *Lkb1*-deficient lung tumors display the

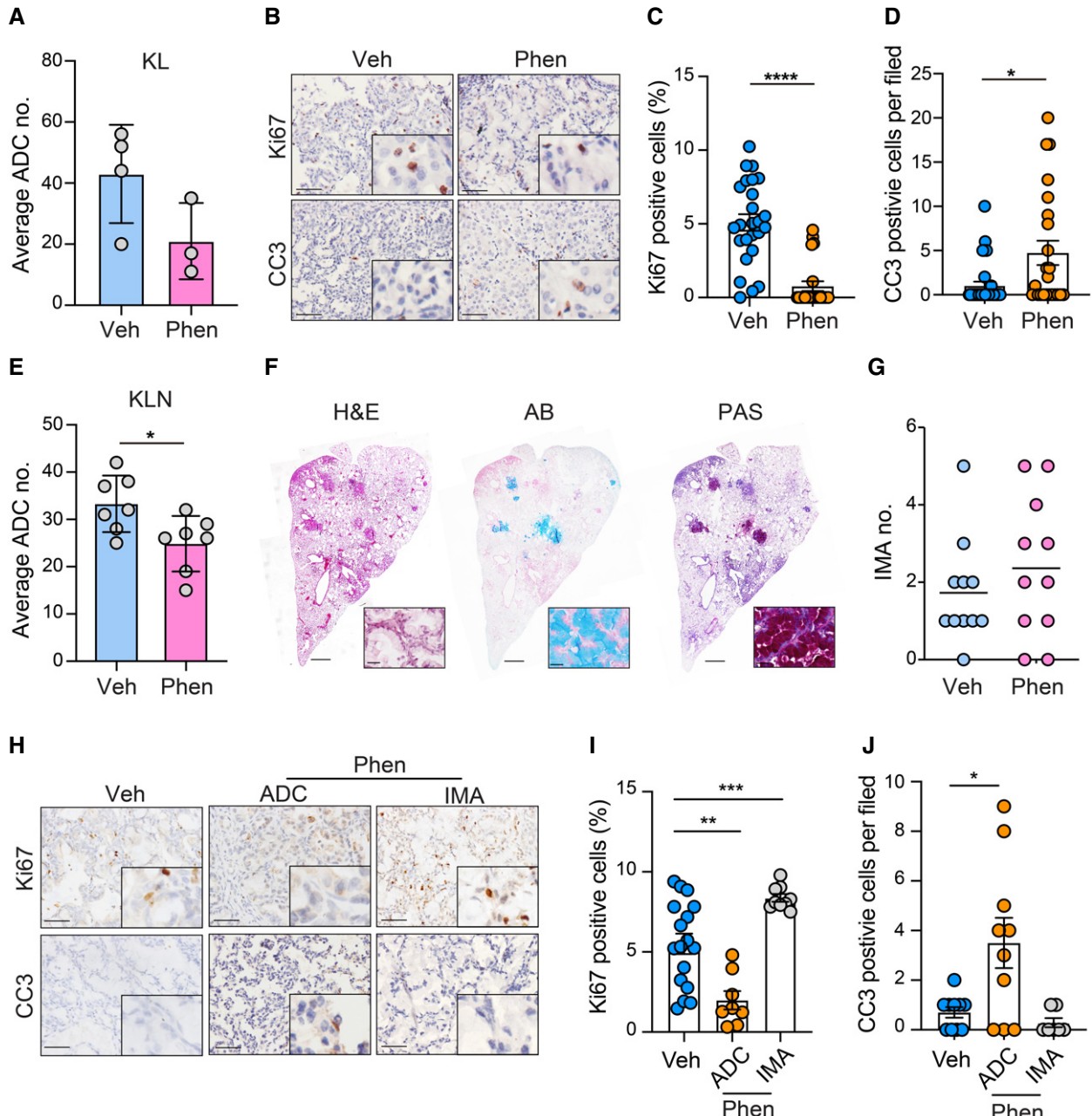

**Figure 5. Mucinous tumors from KLN model are resistant to phenformin treatment.**

A    Statistic analysis of ADC numbers of vehicle (Veh) or phenformin (Phen) treatment group in KL mice (n = 4 in Veh group, n = 3 in Phen group). Results were shown as mean ± SD.

B    IHC staining of Ki67- and CC3-positive cells in lung ADC of vehicle and phenformin (Phen) groups. Scale bar, 50 μm.

C, D    Statistical analysis of Ki67 (C) and CC3 (D) activity in ADC lung sections of vehicle (Veh) and phenformin (Phen) groups. The numbers of tumors analyzed over vehicle and phenformin groups were 25, 22 for Ki67, 32, 23 for CC3. Results were shown as mean ± SEM. Significance was calculated by two-tailed unpaired Student's t-test with Welch's correction. ****P < 0.0001 (C), *P = 0.0162 (D).

E    Statistic analysis of ADC numbers of vehicle (Veh) or phenformin (Phen) treatment group in KLN mice (n = 7 per group). Results were shown as mean ± SD. Significance was calculated by two-tailed unpaired Student's t-test with Welch's correction. *P < 0.05.

F    Representative photographs of H&E, AB and PAS staining in IMA treated with phenformin (Phen). Scale bar, 150 μm. Inset: Representative enlarged photographs of lung lobe of KLN mice treated with phenformin. Scale bar, 50 μm.

G    Statistic analysis of IMA numbers of vehicle or phenformin (Phen) treatment group in KLN mice (n = 11 per group). Results were shown as mean.

H    IHC staining of Ki67- and CC3-positive cells in murine lung ADC and IMA of vehicle and phenformin (Phen) groups. Scale bar, 50 μm.

I, J    Statistical analysis of Ki67 (I) and CC3 (J) activity in ADC and IMA lung sections of vehicle and phenformin (Phen) groups. The numbers of tumors analyzed over vehicle group and phenformin groups for ADC and IMA were 17, 8, 9 for Ki67, 10, 10, 7 for CC3. Results were shown as mean ± SEM. Significance was calculated by one-way ANOVA with Dunnett's multiple comparisons test. **P = 0.0010 (I), ***P = 0.0009 (I), *P = 0.0427(J).

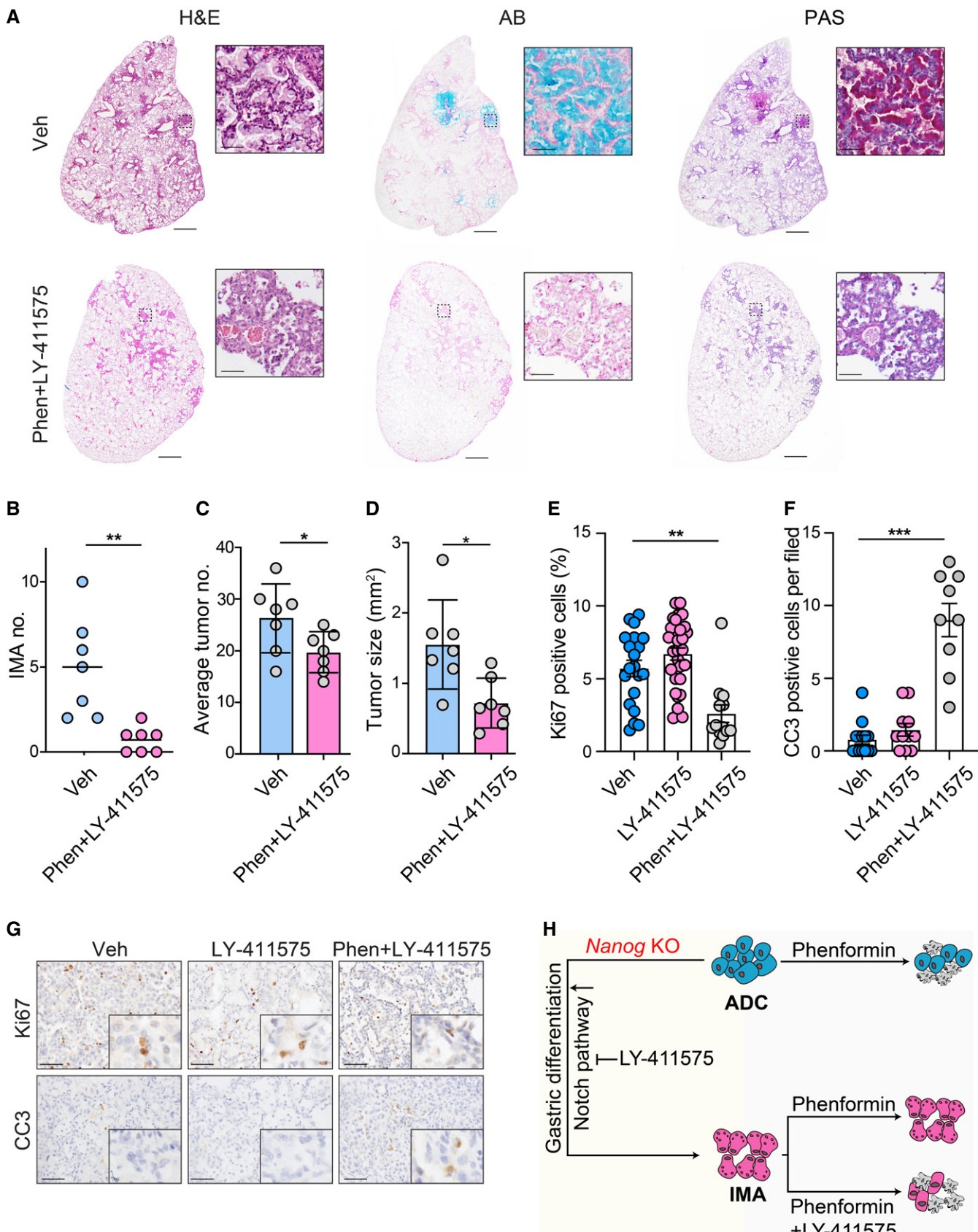

**Figure 6.**

**Figure 6. Concomitant therapy of phenformin and LY411575 suppresses mucinous tumor emergence and overall tumor progression.**

A   H&E, PAS, and AB staining of lung sections in KLN mice with vehicle or a combinational therapy of phenformin (Phen) and LY-411575. Representative images were shown. Scale bar, 150 μm (left panel), 50 μm (right panel).

B–D   Statistical analysis of mucinous tumor incidence (B), total tumor number (C), and tumor size (D) in KLN mice with vehicle or combined treatment (n = 7 per group). Results were shown as mean (B). Results were shown as mean ± SD (C&D). Significance was calculated by two-tailed unpaired Student's *t*-test with Welch's correction. **P = 0.0078 (B), *P = 0.0490 (C), *P = 0.0134 (D).

E, F   Statistical analysis of Ki67 (E) and CC3 (F) reactivity in lung sections of KLN mice following vehicle, LY-411575 treatment or a combinational therapy of phenformin and LY-411575. The numbers of tumors analyzed over vehicle, phenformin, and combinational therapy groups were 20, 30, 13 for Ki67, 14, 11, 9 for CC3. Results were shown as mean ± SEM. Significance was calculated by one-way ANOVA with Dunnett's multiple comparisons test. **P = 0.0014 (E), ***P = 0.0001 (F).

G   IHC staining for Ki67 and CC3 in lung tumor tissues of KLN mice in vehicle, LY-411575 treatment group and combinational group. Representative images were shown. Scale bar, 50 μm.

H   Schematic illustration shows that Nanog is a barrier for KL ADC to IMA transition. Notch signaling is up-regulated in IMA and promotes mucinous differentiation which could be reversed by γ-secretase inhibitor LY-411575. LY-411575 or Phenformin treatment has little effect on mucinous tumor progression. Combination therapy overcomes the treatment resistance imposed by mucinous lineage transition through induction of cell apoptosis.

unique stem cell-like expression pattern, an event not observed in K or KP models. *Nanog*, a master regulator for stem cell maintenance, is significantly up-regulated in KL tumors compared with K or KP tumors. Using GEMM with *Nanog* deletion, we uncover an unexpected mucinous trans-differentiation, which specifically occurs in KL model. These findings underline a context-dependent regulation of the lineage transition program (Maeda *et al*, 2012; Snyder *et al*, 2013; Mollaoglu *et al*, 2018), and *Lkb1* deficiency appears to be a prerequisite to establish tumor plasticity, while additional factors are required to direct specific lineage switching. We identify *Nanog* as a crucial barrier for the mucinous differentiation in *Lkb1*-deficient tumors, which highlights a potential functional link between *Nanog* and *Lkb1* deficiency-mediated cancer stemness and plasticity.

Our data show that *Nanog* deletion initiates the transition from KL ADC to mucinous tumors. This process involves the increased expression of certain stomach-restricted genes such as *Hnf4α, Vsig1, Gkn3, Ctse* and *Onecut2* in *Nanog*-deleted mucinous tumors. Previous reports have shown that gastric differentiation program was triggered in murine lung ADC following NKX2-1 deletion, which involved the interaction of NKX2-1 and FOXA1/ FOXA2 to re-program FOXA1/FOXA2 binding sites and preferential activation of certain target genes (Minoo *et al*, 2007; Gao *et al*, 2008; Wederell *et al*, 2008; Camolotto *et al*, 2018). Some of these genes are normally restricted to the GI tract and undetectable in normal lung epithelia (Menheniott *et al*, 2010). It has been also suggested that NANOG associates with androgen receptor (AR) and FOXA1 to regulate pro-differentiation genes and castration resistance in prostate cancer (Jeter *et al*, 2016). It is currently unknown whether NANOG operates in the signaling complex with NKX2-1/FOXA1/FOXA2 related to the gastric differentiation or if other mechanisms independent of NKX2-1 are engaged. In addition, we currently do not know exactly how *Nanog* is regulated by Lkb1 deficiency. Several studies propose that the absence of *Lkb1* results in STAT3 activation, which further increases the expression of *Oct4* and *Nanog* (Lai *et al*, 2012; Sengupta *et al*, 2017). A recent work also reports that AMPK, the downstream target of LKB1, directly phosphorylates NANOG to promote SPOP-mediated ubiquitination and degradation (Wang *et al*, 2019). These studies indicate that *LKB1* might regulate the transcript as well as protein abundance of *NANOG* through multiple ways. Future work will be interesting to clarify the molecular bases in the regulation of *Nanog* by *Lkb1* during gastric differentiation.

It is worth noting that the expression patterns of NANOG seem different between human and murine lung cancers, e.g., nuclear NANOG expression is observed in mouse model but rarely detectable in human lung cancer specimens. Our data show that NANOG is expressed in KL mouse model and functionally important in maintaining lung cancer stemness; however, it is not significantly expressed in human lung ADC. Differences among species have been previously observed in other cases of lung pathogenesis. For instance, cystic fibrosis is a complex disease with more than one thousand reported mutations in the human *CFTR* gene. Interestingly, mice with different *Cftr* mutants show different phenotypes and affect different organs; however, no single animal model could fully recapitulate all aspects of this disease (Kent *et al*, 1997; Guilbault *et al*, 2007). Another example is severe acute respiratory syndrome coronavirus 2 (SARS-Cov-2), for which unlike in humans, angiotensin-converting enzyme 2 (ACE2) does not support the cellular entry in mice, thus limiting the relevance of mouse studies (Bao *et al*, 2020). Furthermore, in *Kras* mutant lung cancer, Sca-1 positive cells are considered the cells-of-origin in mouse, while the gene is not expressed in human (Kim *et al*, 2005). Similarly here, it will be important to further understand the differences of NANOG expression and function between mouse and human lung cancer pathogenesis for future preclinical studies.

We uncover that Notch signaling is activated during the gastric differentiation. Recent studies have revealed a central role of Notch pathway during pulmonary development and trans-differentiation. Notch signaling activation promotes luminal differentiation of airway basal cells toward goblet secretory lineages (Rock *et al*, 2011) and enables the trans-differentiation of PNECs (pulmonary neuroendocrine cells) to the fate of club cells, ciliated cells, and goblet cells in response to lung injury (Yao *et al*, 2018). These findings are in agreement with the role of Notch signaling that we find here during gastric differentiation of lung cancer. Further studies are required to elucidate the detailed mechanism by which Notch signaling is regulated in IMA.

Targeting lineage-dependent signaling has been shown to be preclinically/clinically beneficial in multiple cancers (Reynolds & Lemons, 2001; Campos *et al*, 2010; Lockwood *et al*, 2012; Rohle *et al*, 2013; Saez-Ayala *et al*, 2013; Shimamura *et al*, 2013; Tremblay *et al*, 2014; Yan *et al*, 2014). Although Notch signaling perturbation represses mucus production, we find that such treatment has little effect on tumorigenesis. Moreover, mucinous tumors are resistant to phenformin treatment, indicating that cell lineage switch emerges as

a protective mechanism for cancer cells to evade therapeutic treatment. We propose a combinational treatment of γ-secretase inhibitor and phenformin, which indeed limits mucinous trans-differentiation as well as tumor progression. Our data describe a novel strategy, which targets lineage-related signaling to enhance therapeutic efficacy of phenformin in *Lkb1*-deficient IMA. Collectively, this study demonstrates that Nanog functions as a critical barrier for an unexpected mucinous differentiation through Notch pathway in KL tumors and proposes the concomitant therapy of LY-411575 and phenformin for *Lkb1*-deficient lung tumors with mucinous differentiation, which might provide new strategy to target human IMA.

# Materials and Methods

## Patients and tissue samples

Patient specimens of 76 IMA and 175 ADC were collected for immune staining. The 76 IMA were obtained from Shanghai Pulmonary Hospital, Tongji University, Shanghai, China from 2007 February to 2013 May with the approval of the ethics committee of the Shanghai Pulmonary Hospital. The 175 human lung ADC were obtained from Shanghai Cancer Center, Fudan University, Shanghai, China and with the approval of the ethics committee of the Shanghai Cancer Center. Tumor types were determined according to WHO classification. All samples were collected with written consent of patients. Informed consent was obtained from all subjects and the experiments conformed to the principles set out in the WMA Declaration of Helsinki and the Department of Health and Human Services Belmont Report.

## Mouse model and treatment

The $Kras^{LSL-G12D/+}$; $P53^{flox/flox}$; and $Lkb1^{flox/flox}$ mice were originally provided by Dr. Tyler Jacks (Cambridge, MA), Dr. Ranold DePinho (Boston, MA) respectively. $Nanog^{flox/flox}$ mice were generated by Dr. Lei Xiao (Zhejiang University, China). The cohorts were crossed with $Nanog^{flox/flox}$ mice respectively to generate the $Kras^{LSL-G12D/+}$; $Nanog^{flox/flox}$, $Kras^{LSL-G12D/+}$; $P53^{flox/flox}$; $Nanog^{flox/flox}$ and $Kras^{LSL-G12D/+}$; $Lkb1^{flox/flox}$; $Nanog^{flox/flox}$ mice. All mice were kept in specific pathogen-free environment of Shanghai Institute of Biochemistry and Cell Biology, received humane care and treated in strict accordance with protocols approved by the Institutional Animal Care and Use Committee of the Shanghai Institutes for Biological Sciences, Chinese Academy of Sciences. Age- and sex-matched animals were randomly grouped. All mice used in this study were from mixed gender. Mice were treated with Ad-Cre virus at $2 \times 10^6$ PFU by nasal inhalation at 7–8 weeks of age. Mice were sacrificed at 4–8 weeks post-initiation for gross inspection and histopathological examination. Tumor numbers were counted under microscope, and tumor burden was determined through Photoshop and Image J.

For drug efficacy evaluation, LY-411575 (Sigma, SML0506) was formulated in 1% carboxymethyl cellulose (CMC), 0.25% Tween 80. 8 weeks after tumor initiation, KLN mice were given LY-411575 (3 mg/kg/day) via gavage. Phenformin (Sigma P7045) was administered in drinking water (at 1.8 mg/ml). After treatment, mice were sacrificed for detailed pathological analysis.

## Histological examination

Mice lung lobes were inflated with formalin, fixed overnight and dehydrated in ethanol, embedded in paraffin, sectioned (5μm) followed by staining with hematoxylin and eosin (Sigma).

For IHC staining, slides were de-paraffinized in xylene and ethanol, and rehydrated in water. Slides were quenched in hydrogen peroxide (3%) to block endogenous peroxidase activity. Antigen retrieval was performed by heating slides in a microwave for 20 min in sodium citrate buffer (pH 6.0). The primary antibodies were incubated at 4°C overnight followed by using the SPlink Detection Kits (Biotin-Streptavidin HRP Detection Systems) according to the manufacturer's instructions. IHC scoring of NANOG and NICD expression was performed as previously described (Gao *et al*, 2014). Briefly, the score was determined as follows: Expression of the above genes in over 75% of the tumor cells was scored as high; expression in 10–75% of the tumor cells was scored as medium; expression in < 10% of the tumor cells was scored as low. Percentage of positive cells was calculated by the ration of IHC-stained positive cells to all tumor cells in an individual tumor.

Paraffin-embedded lung tissues were incubated with following antibodies: anti-NANOG (Bethyl, A300-397A, 1:500 dilution), anti-NANOG (Cell Signaling Technologies, 4903, 1:1,000 dilution), anti-DNp63 (Abcam, ab124762, 1:20,000 dilution), anti-NICD (Abcam, ab8925, 1:500 dilution), anti-HES1 (Santa cruz, sc-25392, 1:1,000 dilution), anti-HES5 (Abcam, ab194111, 1:1,000 dilution), anti-HEY1 (Abcam, ab22614, 1:1,000 dilution), anti-SP-C (Chemicon, AB3786, 1:2,500 dilution), anti-CC10 (Santa Cruz, sc-365992, 1:200 dilution), anti-HNF4α (Sangon Biotech, D164196, 1:500 dilution), anti-GKN1 (Abclonal, A13107, 1:500 dilution), anti-MUC5AC (Abcam, ab3649, 1:500 dilution), anti-CK20 (Sangon Biotech, D162650, 1:500 dilution), anti-NKX2-1 (Abcam, ab133638,1:500 dilution), anti-LKB1 (Cell Signaling Technologies, 13031, 1:300 dilution), anti-Ki-67 (Novocastra, NCL-Ki67P, 1:500 dilution), and anti-cleaved caspase-3 (Cell Signaling Technologies, 9661, 1:800 dilution).

For Alcian Blue and PAS staining, tumor sections were stained with 1% Alcian Blue pH 2.5 and Periodic acid–Schiff reagent (Polyscientific Company) according to the manufacturer's instructions.

## Quantitative real-time PCR

Total RNA was extracted using Trizol reagent (Invitrogen) and phenol/chloroform methods. RNA was retro-transcribed into first-strand complementary DNA using RevertAid First-Strand cDNA Synthesis Kit (Fermentas). cDNAs were subjected to quantitative real-time PCR with gene-specific primers on 7,500 Fast Real-Time PCR System (Applied Biosystems) using SYBR-Green Master PCR mix (Invitrogen). Sequences of real-time PCR primers are listed in Table EV2. RNA of KL and KLN ADC tumors was extracted, and cDNAs were performed by real-time PCR using *Muc5ac* primers to distinguish tumors into mucinous tumors or non-mucinous tumors.

## Plasmid constructs and transfection

Full-length human NANOG cDNA cloned into pENTER vector was purchased from Vigene Biosciences. Cells were seeded in 6-well plates and transfected with the plasmid using Lipofectamine 2000

(Invitrogen) according to the manufacturer's instruction. The amount of transfected DNA was kept constant by addition of corresponding amounts of the backbone plasmid. After 48 h of incubation, whole-cell lysates were collected.

## Western blot

Tumors were homogenized using Percellys 24 homogenizer (Bertin). Homogenized tumors or whole-cell lysates of ADC cell lines were prepared in loading buffer (10% SDS, 1 mM DTT and glycerin) and incubated at 100°C for 10 min. Equal volumes of proteins were resolved by SDS–PAGE and transferred onto PVDF membranes. After incubation in blocking buffer (50 mM Tris-buffered saline [pH 7.4] containing 5% non-fat dry milk and 0.1% Tween-20), the membranes were probed with the primary antibodies, followed by incubation with HRP-linked goat anti-rabbit IgG (CST, 7074S, 1:5000 dilution). Bands were revealed with an ECL kit (Thermo Fisher Scientific) prior to detection on SAGECREATION (Sage Creation Science Co, Beijing).

## RNA-seq library construction and sequencing

Total RNA was extracted from $Kras^{LSL-G12D/+}$, $Kras^{LSL-G12D/+}$; $P53^{flox/flox}$ and $Kras^{LSL-G12D/+}$; $Lkb1^{flox/flox}$ mice tumors (3 samples/mouse model) using Trizol reagent (Invitrogen) and phenol/chloroform methods. Approximately 2μg of RNA from each sample was used to generate RNA-seq cDNA libraries for sequencing using the TruSeq RNA Sample Prep Kit v2 (Illumina, Inc., San Diego, CA). Sample preparation followed the manufacturer's protocol. The amplified cDNA fragments were analyzed using the 2100 Bioanalyzer (Agilent Technologies, Inc., Santa Clara, CA) to determine fragment quality and size. Library concentrations were determined by Qubit Fluorometric Quantitiation (Life Technologies Corporation, Carlsbad, CA). Sequencing of 280bp paired-end reads was performed with an Illumina NovaSeq 6000 instrument at the Berry Genomics (Shanghai, CN).

## RNA-seq data analysis

Data have been submitted to OEP000813 (https://www.biosino.org) (for K and KL tumors), RNA-seq datasets of KP tumors were obtained from the SRA archive (SRP158744) (Yao *et al*, 2019). Gene expression profile was generated by TopHat (v2.1.1) and HTseq (v0.6.0). Differential expression genes (log2 fold change > 1, *P* value < 0.05) were selected after the processing of DESeq (R package). Heatmap was drawn by R (v3.4.2) using part of genes up-regulated in KL tumor samples. GSEA (v3.0) from Broad Institute Platform was perform to pathway enrichment and *P* value < 0.05 was set as statistical significance. Characteristic gene sets were analyzed according to the genes presenting the strongest enrichment scores for each gene set.

## Statistical analyses

For comparing means of 2 groups, two-tailed unpaired Student's *t*-test with Welch's correction was used. For comparing means of 3 groups, significance was determined using one-way ANOVA with Dunnett's multiple comparisons test. Significance of grade of IHC staining was calculated by χ2 test for trend. Student's *t*-test and one-way ANOVA analyses were performed by Prism GraphPad software. χ2 test was performed by R package (library ("coin")). Differences

### The paper explained

**Problem**

Lineage plasticity has emerged as a mechanism allowing evasion in response to targeted therapies in several cancer types including non-small cell lung cancer (NSCLC). Recent studies have revealed an unexpected gastric differentiation program of lung adenocarcinoma, mimicking human invasive mucinous adenocarcinoma (IMA) with expression of gastric markers. However, the pathogenesis of IMA remains largely unknown. Moreover, the effect of this lineage switching on the therapeutic response is unclear.

**Results**

Our study reveals that *Lkb1*-deficient *Kras* tumors show significant up-regulation of Nanog expression, and knockout of Nanog in $Kras^{LSL-G12D/+}$; $Lkb1^{flox/flox}$ (KL) tumors promotes gastric differentiation and occurrence of IMA. Importantly, we find that low or negative expression of NANOG or LKB1 is concurrent in about 89% of human IMA. We demonstrate that Notch signaling inactivation blocks mucin production in mice. Notably, this lineage switching desensitizes *Lkb1*-deficient lung tumors to phenformin treatment, which selectively targets KL tumors. Finally, we find that combinational treatment of γ-secretase inhibitor LY411575 and phenformin is sufficient to prevent mucinous differentiation as well as malignant progression in mice.

**Impact**

These results shed light on the gastric differentiation program involving the Nanog-Notch axis in an *Lkb1*-deficient context. They further provide novel therapeutic insights into human IMA treatment.

with *P* < 0.05 were considered statistically significant. Data were all represented as mean ± SD or mean ± SEM.

# Data availability

The raw RNA-seq data are deposited at https://www.biosino.org/node/project/detail/OEP000813.

**Expanded View** for this article is available online.

# Acknowledgements

We thank Drs. T. Jacks, R. Depinho for providing the mice. We are grateful to Dr. Luonan Chen, Zhonglin Jiang, and Dr. Zhaoyuan Fang for providing bioinformatic supports. We thank Drs. Kwok-Kin Wong, Daming Gao, Yong Chen, Lu Cheng, Xiangkun Han, Yonglong Zhang, Liang Hu, Qibiao Wu, Fei Li, Min Chen, Shun Yao and Hongjun Li for helpful suggestions and technical supports. This work was supported by the National Basic Research Program of China (Grant 2017YFA0505501 to H.J.), Strategic Priority Research Program of the Chinese Academy of Sciences (XDB19020201 to H.J.), National Natural Science Foundation of China (31621003 to H.J., 81872312 to H.J., 91731314 to H.J., 81430066 to H.J., 81871875 to L.H., 81802279 to H.H., 81902326 to X.W.), Basic Frontier Scientific Research Program of Chinese Academy of Science (ZDBS-LY-SM006 to H.J.), State Key Laboratory of Oncogenes and Related Genes (KF-20-03 to H.J.) and the China Postdoctoral Science Foundation (2016M601667 to H.H.).

# Author contributions

HJ and XT conceived the ideas and designed the experiments, acquired the data and performed the analysis as well as interpretation. XT performed the

experiments including mice analyses, clinical data analyses, and biochemical assays. LX generated the Nanog conditional knockout mice. XZ, PZ provided clinical samples and related information. RW, HC provided clinical information. YY performed the bioinformatic analysis. YX, YC, WG provided technical supports. YG, WZ provided supports and suggestions. XT, and HJ wrote the manuscript.

## Conflict of interest

The authors declare that they have no conflict of interest.

## For more information

https://www.ebi.ac.uk/gxa/home; https://www.cbioportal.org

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
