## [Review Process File · EMBO Molecular Medicine]

***Nanog* maintains stemness of *Lkb1*-deficient lung adenocarcinoma and prevents gastric differentiation**

Xinyuan Tong, Yueqing Chen, Xinsheng Zhu, Yi Ye, Yun Xue, Rui Wang, Yijun Gao, Wenjing Zhang, Weiqiang Gao, Lei Xiao, Haiquan Chen, Peng Zhang, Hongbin Ji

DOI: 10.15252/emmm.202012627

Corresponding authors: Hongbin Ji (hbji@sibcb.ac.cn) , Peng Zhang (zhangpeng1121@tongji.edu.cn)

Review Timeline:

Submission Date:	29th Apr 20
Editorial Decision:	14th May 20
Revision Received:	13th Aug 20
Editorial Decision:	28th Aug 20
Revision Received:	5th Nov 20
Editorial Decision:	16th Nov 20
Revision Received:	25th Nov 20
Accepted:	4th Dec 20

Editor: Lise Roth

Transaction Report:

14th May 2020

Dear Dr. Ji,

Thank you for submitting your work to EMBO Molecular Medicine. We have now heard back from the three referees who agreed to evaluate your manuscript. As you will see below, the reviewers raise substantial concerns on your work, which unfortunately preclude its publication in its current form.

The reviewers find that the question addressed by the study is of potential interest, and particularly appreciate the generation of the NANOG conditional knock-out mouse model. However they remain unconvinced by the relevance of the findings for human lung cancer due in part to the uncertain NANOG expression. Human relevance is essential for publication in EMBO Molecular Medicine, and therefore, further consideration in our journal will require a clear demonstration that the results highlighted here are relevant for human lung cancer.

If you feel you can satisfactorily address this point and those listed by the referees, you may wish to submit a revised version of your manuscript. Please attach a covering letter giving details of the way in which you have handled each of the points raised by the referees. A revised manuscript will once again be subject to review and we cannot guarantee at this stage that the eventual outcome will be favorable.

EMBO Molecular Medicine encourages a single round of revision only and therefore, acceptance or rejection of the manuscript will depend on the completeness of your responses included in the next, final version of the manuscript. Should you find that the requested revisions are not feasible within the constraints outlined here and prefer, therefore, to submit your paper elsewhere, we would welcome a message to this effect.

When submitting your revised manuscript, please carefully review the instructions that follow below. Failure to include requested items will delay the evaluation of your revision:

- 1) A .docx formatted version of the manuscript text (including legends for main figures, EV figures and tables). Please make sure that the changes are highlighted to be clearly visible.
- 2) Individual production quality figure files as .eps, .tif, .jpg (one file per figure).
- 3) A .docx formatted letter INCLUDING the reviewers' reports and your detailed point-by-point responses to their comments. As part of the EMBO Press transparent editorial process, the point-by-point response is part of the Review Process File (RPF), which will be published alongside your paper.
- 4) A complete author checklist, which you can download from our author guidelines (<https://www.embopress.org/page/journal/17574684/authorguide#submissionofrevisions>). Please

insert information in the checklist that is also reflected in the manuscript. The completed author checklist will also be part of the RPF.

6) Before submitting your revision, primary datasets produced in this study need to be deposited in an appropriate public database (see <https://www.embopress.org/page/journal/17574684/authorguide#dataavailability>). Please remember to provide a reviewer password if the datasets are not yet public. The accession numbers and database should be listed in a formal "Data Availability " section (placed after Materials & Method). Please note that the Data Availability Section is restricted to new primary data that are part of this study.

7) We would also encourage you to include the source data for figure panels that show essential data. Numerical data should be provided as individual .xls or .csv files (including a tab describing the data). For blots or microscopy, uncropped images should be submitted (using a zip archive if multiple images need to be supplied for one panel). Additional information on source data and instruction on how to label the files are available at .

8) Our journal encourages inclusion of *data citations in the reference list* to directly cite datasets that were re-used and obtained from public databases. Data citations in the article text are distinct from normal bibliographical citations and should directly link to the database records from which the data can be accessed. In the main text, data citations are formatted as follows: "Data ref: Smith et al, 2001" or "Data ref: NCBI Sequence Read Archive PRJNA342805, 2017". In the Reference list, data citations must be labeled with "[DATASET]". A data reference must provide the database name, accession number/identifiers and a resolvable link to the landing page from which the data can be accessed at the end of the reference. Further instructions are available at .

9) We replaced Supplementary Information with Expanded View (EV) Figures and Tables that are collapsible/expandable online. A maximum of 5 EV Figures can be typeset. EV Figures should be cited as 'Figure EV1, Figure EV2' etc... in the text and their respective legends should be included in the main text after the legends of regular figures.

- Additional Tables/Datasets should be labeled and referred to as Table EV1, Dataset EV1, etc. Legends have to be provided in a separate tab in case of .xls files. Alternatively, the legend can be supplied as a separate text file (README) and zipped together with the Table/Dataset file. See detailed instructions here: .

10) The paper explained: EMBO Molecular Medicine articles are accompanied by a summary of the articles to emphasize the major findings in the paper and their medical implications for the non-

specialist reader. Please provide a draft summary of your article highlighting

11) For more information: There is space at the end of each article to list relevant web links for further consultation by our readers. Could you identify some relevant ones and provide such information as well? Some examples are patient associations, relevant databases, OMIM/proteins/genes links, author's websites, etc...

12) Every published paper now includes a 'Synopsis' to further enhance discoverability. Synopses are displayed on the journal webpage and are freely accessible to all readers. They include a short stand first (maximum of 300 characters, including space) as well as 2-5 one-sentences bullet points that summarizes the paper. Please write the bullet points to summarize the key NEW findings. They should be designed to be complementary to the abstract - i.e. not repeat the same text. We encourage inclusion of key acronyms and quantitative information (maximum of 30 words / bullet point). Please use the passive voice. Please attach these in a separate file or send them by email, we will incorporate them accordingly.

Please also suggest a striking image or visual abstract to illustrate your article. If you do please provide a png file 550 px-wide x 400-px high.

13) As part of the EMBO Publications transparent editorial process initiative (see our Editorial at <http://embomolmed.embopress.org/content/2/9/329>), EMBO Molecular Medicine will publish online a Review Process File (RPF) to accompany accepted manuscripts.

In the event of acceptance, this file will be published in conjunction with your paper and will include the anonymous referee reports, your point-by-point response and all pertinent correspondence relating to the manuscript. Let us know whether you agree with the publication of the RPF and as here, if you want to remove or not any figures from it prior to publication.

I look forward to receiving your revised manuscript.

Yours sincerely,

Lise Roth

Lise Roth, PhD
Editor
EMBO Molecular Medicine

To submit your manuscript, please follow this link:

Link Not Available

*Additional important information regarding figures and illustrations can be found at <http://bit.ly/EMBOPressFigurePreparationGuideline>

***** Reviewer's comments *****

Referee #1 (Remarks for Author):

Comments

Tong et al produced a mucinous lung tumor mouse model by expressing mutant Kras while deleting both Lkb1 and Nanog in the lung. They further treated the model mice with LY-411575 and/or phenformin. The phenotype of this mouse model is very interesting, and the effect of the drugs is striking. However, there are some points that should be addressed.

(1) The authors showed genes (SMAD5, TBX3, NANOG, SMAD4, MYC, JARID2, MEIS1 and SETDB1) in Figure 1C that are highly expressed in the KL mice compared to those in the K or KP mice. According to TCGA dataset from the Human Protein Atlas, SMAD5, TBX3, SMAD4, MYC, JARID2, MEIS1 and SETDB1 are highly expressed in a portion of human lung cancer, but NANOG is not. NANOG is highly expressed only in human testis cancer.

The authors themselves did immunohistochemistry using human lung cancer specimens with anti-NANOG antibody (Bethyl, A300-397A) to show that NANOG is expressed in ~40% of human lung adenocarcinoma (ADC) and ~8% of IMA (medium to high expression; Figure 3). The authors cite these papers below to support their claim:

Du Y, Ma C, Wang Z, Liu Z, Liu H, Wang T (2013) Nanog, a novel prognostic marker for lung cancer. *Surg Oncol* 22: 224-9

Zhao L, Liu J, Chen S, Fang C, Zhang X, Luo Z (2018) Prognostic significance of NANOG expression in solid tumors: a meta-analysis. *Onco Targets Ther* 11: 5515-5526

The authors need to address this discrepancy between the datasets in the public domain (e.g., TCGA) and their data plus Du et al (2013) since the expression of NANOG in lung cancer does not seem to be well accepted in the lung cancer field.

And regarding NANOG antibodies,

the authors use Bethyl, A300-397A. According to the Bethyl website, the reactivity of this antibody is mouse. Since the protein identity % is 56% between human and mouse (please confirm this if it's correct), the authors need to confirm by western blotting if this antibody detects human NANOG.

Du et al (2013) used a NANOG antibody (1E6C4) from Cell Signaling. According to the Cell Signaling website, it reacts with human NANOG. However, this antibody detected NANOG in the cytoplasm but not in the nucleus of the lung cancer cells according to their representative images (see Du paper). Is there any possibility that this antibody is more suitable to look at the expression of NANOG in human lung cancer tissues than the Bethyl antibody?

Zhao et al (2018) did not produce any primary data by themselves since they showed data of a meta-analysis study. It would be good to cite a paper of a study which conducted original research looking at the expression of NANOG in human lung cancer.

(2) And the authors used human A549 lung cancer cells in their previous paper (Han et al 2013). According to the cell line database at the Human Protein Atlas, A549 cells do not express NANOG. Is there any human lung cancer cell line that expresses NANOG at a considerable level? The authors should show the levels of expression using mRNA and protein extracted from the cell lines if they have any positive data. This will support the authors' claim that NANOG is highly expressed in human lung cancer.

(3) Is there a possibility that NANOG may be important in mouse lung cancer but not in human lung cancer? If so, it might be good to investigate the mechanism of the difference.

(4) Is there any mechanistic study? The data shown by the authors indicate that NANOG inhibits the expression of genes related to gastric differentiation. How does NANOG inhibit these genes? What's the molecular mechanism? The authors may be able to identify the mechanism in vitro.

(5) Is HES1, HES5, HEY1 or HEY2 expressed in human IMA? If so, please show the expression level in Figure 3D.

(6) What's the expression of NKX2-1 in the KLN mouse lungs treated with LY-411575?

(7) Did LY-411575 influence airway cilia?

(8) It would be good to show low magnified images of AB and PAS as well as H&E in Figures 5B and 6A (only H&E image is shown now), which would help the readers to get an idea about the degree of prevalence of IMA in the mouse lungs.

(9) Winslow et al (2011) should be cited in this part of the Introduction:

The transcription factor NKX2-1 generally expresses in human lung adenocarcinoma (ADC) but decreases with poor differentiation and malignant progression (Boland et al, 2018; Cai et al, 2014; Cha & Shim, 2017).

(10) Bar graphs with dot plots are recommended to ensure data integrity.

Referee #2 (Comments on Novelty/Model System for Author):

1. RNA-seq data showed that stem cell pluripotency expression pattern was significantly enriched in KL tumors in contrast to K or KP tumors (Fig 1A and B). These data are expectable because KL tumor exhibits strong plasticity and lineage transition from adenocarcinoma (ADC) to squamous cell carcinoma (SCC) in previous literatures.
2. Generation of Nanog conditional knockout mice which show mucinous differentiation is a great work!
3. Combination therapy with phenformin and Notch inhibitor LY-411575 has anti-tumor effect in KLN mice through inhibiting mucinous lineage transition. However, clinical application of this combination therapy in human IMA, which cancer cells already existed, is doubtful.

Referee #2 (Remarks for Author):

This manuscript by Xinyuan Tong et al. used Kras-driven lung cancer genetically engineered mouse model (GEMM) to investigate an undruggable subtype, invasive mucinous adenocarcinoma (IMA). They demonstrated that Nanog deletion in LKB1 deficiency lung adenocarcinoma mouse model promoted mucinous differentiation of tumor, which could be blocked by Notch inhibitor, LY-411575. They also showed the frequently concurrent loss of NANOG and LKB1 expression in human IMA. Finally, they displayed that IMA was more resistant to phenformin treatment, an analog of anti-diabetes drug metformin, compared to that in adenocarcinoma. This resistance could be overcome by LY-411575. The merits of this study are GEMM and abundant human lung cancer tissue. However, I still have some concerns about experiments.

Major comments:

1. According to the result of "Nanog deletion in KL tumors triggers gastric differentiation", the expression of LKB1 and Nanog should be investigated in human adenocarcinoma and IMA. Only Nanog (Fig.3C) in human lung cancer tissue is not enough.
2. To investigate the different resistance of mucinous tumor and adenocarcinoma to phenformin (which can selectively target LKB1-deficient ADC), phenformin should be used in KL and KLN mice. All lung tumors in KL mice supposed to be LKB1-deficient ADC and all tumors in KLN mice supposed to be LKB1 and Nanog deficient. I wondered the experimental design in Fig5. Moreover, author state "little effect was observed on the occurrence of mucinous tumors after phenformin treatment (Fig 5B and C)", but the result in Fig.5C seems not! The mice number of Fig.5C (5 vs. 7, vehicle vs. phenformin) should be equal.

Referee #3 (Remarks for Author):

The study by Tong et al titled Nanog maintains stemness of Lkb1-deficient lung adenocarcinoma and prevents gastric differentiation utilizes GEMM models to further characterize the genetic components involved in invasive mucinous adenocarcinoma development. This manuscript is well written and provides a number of novel findings including the role of Nanog.

I do however have some concerns and these are listed below

- I know the numbers aren't statistically significant but it appears as if KLN mice have more small tumors (increased tumor number but decreased tumor burden). I'm not sure if this is true or the significance of this observation if it is true but the authors might want to evaluate this further.
- Some more information about the IMA tumors would be appreciated. Are they actually invasive?

Do they metastasize? A more detailed description of the tumor features, as described by a pathologist, should be provided.

- I'm not sure I would agree with the statement on page 8 that "We found that the mucinous ADC closely recapitulated the histological characteristics of human lung IMA". From the information presented, there are some similarities but also a lot of structural differences. I think additional measures or description of the shared histological features are required to support this claim.
- For the qRT-PCR analyses of the tumors (ie Fig 2F), how was the RNA extracted? Was laser capture microdissection or some sort of sorting system used? If bulk lungs were used, how did the authors select the lungs to ensure parameters like similar tumor burden, enrichment for ADC or IMA in the samples used, etc.
- Only 5 of the 9 gastric differentiation genes were upregulated. So, what does this mean when almost half of the gastric differentiation genes were not significantly upregulated? Also, why did the applicants use GKN1 in the immunohistochemistry when this marker was not upregulated at the mRNA level?
- Unfortunately, statistics is not an area of expertise for me however, I think the statistics in this paper should be evaluated more closely. For example, a Student's t-test is not a suitable statistical test for comparing means of 3 groups such as the data in Fig 2C, Fig 5E,F, Fig 6F,G. Also given the large error bars in Fig 4C,D I find it improbable that these means are statistically different unless a very large number of samples were used. More details on how these counts were performed and analyzed, should be provided.

Re: EMM-2020-12627

Nanog maintains stemness of *Lkb1*-deficient lung adenocarcinoma and prevents gastric differentiation

Referee #1 (Remarks for Author):

Comments

Tong et al produced a mucinous lung tumor mouse model by expressing mutant Kras while deleting both Lkb1 and Nanog in the lung. They further treated the model mice with LY-411575 and/or phenformin. The phenotype of this mouse model is very interesting, and the effect of the drugs is striking. However, there are some points that should be addressed. (1) The authors showed genes (SMAD5, TBX3, NANOG, SMAD4, MYC, JARID2, MEIS1 and SETDB1) in Figure 1C that are highly expressed in the KL mice compared to those in the K or KP mice. According to TCGA dataset from the Human Protein Atlas, SMAD5, TBX3, SMAD4, MYC, JARID2, MEIS1 and SETDB1 are highly expressed in a portion of human lung cancer, but NANOG is not. NANOG is highly expressed only in human testis cancer.

The authors themselves did immunohistochemistry using human lung cancer specimens with anti-NANOG antibody (Bethyl, A300-397A) to show that NANOG is expressed in ~40% of human lung adenocarcinoma (ADC) and ~8% of IMA (medium to high expression; Figure 3). The authors cite these papers below to support their claim:

Du Y, Ma C, Wang Z, Liu Z, Liu H, Wang T (2013) Nanog, a novel prognostic marker for lung cancer. Surg Oncol 22: 224-9

Zhao L, Liu J, Chen S, Fang C, Zhang X, Luo Z (2018) Prognostic significance of NANOG expression in solid tumors: a meta-analysis. Onco Targets Ther 11: 5515-5526

The authors need to address this discrepancy between the datasets in the public domain (e.g., TCGA) and their data plus Du et al (2013) since the expression of NANOG in lung cancer does not seem to be well accepted in the lung cancer field.

And regarding NANOG antibodies, the authors use Bethyl, A300-397A. According to the Bethyl website, the reactivity of this antibody is mouse. Since the protein identity % is 56% between human and mouse (please confirm this if it's correct), the authors need to confirm by western blotting if this antibody detects human NANOG.

Du et al (2013) used a NANOG antibody (1E6C4) from Cell Signaling. According to the Cell Signaling website, it reacts with human NANOG. However, this antibody detected NANOG in the cytoplasm but not in the nucleus of the lung cancer cells according to their representative images (see Du paper). Is there any possibility that this antibody is more suitable to look at the expression of NANOG in human lung cancer tissues than the Bethyl antibody?

Zhao *et al* (2018) did not produce any primary data by themselves since they showed data of a meta-analysis study. It would be good to cite a paper of a study which conducted original research looking at the expression of *NANOG* in human lung cancer.

Response: Thanks the reviewer for pointing out this question. We agree with the reviewer that *NANOG* is not highly expressed in most cancer types from the TCGA dataset. At the same time, multiple studies have also shown that *NANOG* was expressed in human lung cancer at RNA or protein level (Watanabe, 2009; Li *et al*, 2013a; Du *et al*, 2013; Li *et al*, 2013b; Park *et al*, 2016; Vaira *et al*, 2013; Liu *et al*, 2014). We have now evaluated *NANOG* mRNA expression using public RNA-Seq dataset from Chinese lung ADC (Chen *et al*, 2019) (Referee Figure 1). We observed certain degree albeit low *NANOG* expression in part of human samples (44/197, 22%; TPM range (0.14, 2.1]). Besides mRNA level analyses, multiple IHC staining studies show that *NANOG* is expressed in 20~70% of human lung ADC (Li *et al*, 2013a; Park *et al*, 2016; Li *et al*, 2013b; Liu *et al*, 2014). Park *et al* used 368 patient samples from Seoul National University Bundang Hospital (Park *et al*, 2016). Liu *et al* used tissue microarray purchased from US Biomax Inc (Liu *et al*, 2014). Li *et al* analyzed 112 human samples from Guilin Medical University Hospital, China (Li *et al*, 2013a). All these studies find that *NANOG* expression significantly correlates with high tumor grade, suggesting the heterogeneity of *NANOG* expression among tumors. Based on these data and reports, we hope the reviewer agree that *NANOG* is expressed in part of human lung cancer at mRNA or protein level and its expression level might not be that high but positively associate with lung tumor grade.

Referee Figure 1. Different *NANOG* expression in human lung ADC. TPM means Transcripts Per Kilobase of exon model per Million mapped reads. Sample numbers were shown on the top of the bar within indicated range of TPM.

The reviewer is right about the NANOG protein identity between human and mouse. NANOG is composed of 305 amino acids including a central homeodomain. Human NANOG polypeptide shares about 58% identity to murine NANOG and the identity of the homeodomain is about 87% (Oh et al, 2005).

Regarding NANOG antibodies, we have now evaluated NANOG expression in human samples using two antibodies: the Bethyl antibody and CST antibody (NANOG (D73G4), 4903). We found that both antibodies detected NANOG protein at similar location (Referee Figure 2A and B). NANOG exists two isoforms with molecular weight 34.4 kDa and 31.9 kDa. There are post-translational modifications affecting the band size of NANOG (Ambady et al, 2010; Ramakrishna et al, 2014). Based on manufactures' instructions and representative photos of western blotting, the CST antibody detects NANOG in the range of 30kDa to 50kDa whereas the Bethyl antibody detects NANOG at around 42kDa. We have now tried both antibodies and found the Bethyl antibody showed some nonspecific bands (Referee Figure 2).

Referee Figure 2. (A-B) Western blot for NANOG and β -TUBULIN in human lung tissues. Bethyl antibody (A) and CST antibody (B) was used to evaluate NANOG expression. Murine embryonic stem cells (ES) was used as the control. N, normal lung. T, human lung tumor.

To test our hypothesis, we then chose the CST antibody to re-perform IHC staining in human ADC and IMA tissues. Due to the limited number of remaining tissue sections, we only examined 76 human IMA tissues. The data showed that NANOG is expressed in around 20% of human lung ADC and barely detectable in human IMA samples (new Figure 3A-C), consistent with our previous findings. We also observed that this antibody detected NANOG in cytoplasm. Previous IHC staining has shown that NANOG protein could be detected in cytoplasm or nucleus in tumors (Li et al, 2013a, 2013b). We speculated that the location of NANOG in tumor cells might depend on tumor grade and the degree of differentiation. To solidify our conclusion, we replaced the original data about human NANOG with new results using the CST antibody (new Figure 3A-C, Figure 3G).

As suggested, we have cited more papers which detect NANOG expression in human lung cancer (See lines 111-112).

(2) And the authors used human A549 lung cancer cells in their previous paper (Han et al 2013). According to the cell line database at the Human Protein Atlas, A549 cells do not express NANOG. Is there any human lung cancer cell line that expresses NANOG at a considerable level? The authors should show the levels of expression using mRNA and protein extracted from the cell lines if they have any positive data. This will support the authors' claim that NANOG is highly expressed in human lung cancer.

Response: Thanks the reviewer for this comment and suggestion. We determined NANOG expression in human lung ADC cell lines using the CST antibody. Several papers have shown that this antibody can detect NANOG expression in lung cancer cell lines including A549 and H1299 (Li et al, 2019; Zhu et al, 2017; Guo et al, 2017). We used the classical method (method 1: lysate cells using RIPA lysis buffer with PMSF) to collect proteins from cell lines. However, we couldn't detect NANOG expression in most human cancer cell lines (Referee Figure 3A). We then tried another method (method 2: lysate cells using 1x Loading lysis buffer and sonicate after collection) which could detect proteins at moderate or even low level. In fact, we found that NANOG was expressed in several human ADC cell lines (Referee Figure 3B). We also used 4 pairs of primers to detect human *Nanog* expression in these cell lines (Referee Figure 3C) (Primer 1, Forward: 5'-ATGCCTGTGATTTGTGGGCC-3' Reverse: 5'-GCCAGTTGTTTTCTGCCAC-3'; Primer 2, Forward: 5'-TTTGTGGGCCTGAAGAAACT-3' Reverse: 5'-AGGGCTGTCCTGAATAAGCAG-3'; Primer 3, Forward: 5'-AAGGTCCCGGTCAAGAACAG-3' Reverse: 5'-CTTCTGCGTCACACCATTGC-3'; Primer 4, Forward: 5'-CCCCAGCCTTTACTCTTCCTA-3' Reverse: 5'-CCAGGTTGAATTGTTCCAGGTC-3'). These results showed that *Nanog* was expressed in some human cancer cell lines.

Referee Figure 3. (A-B) Western blot for NANOG and β -TUBULIN in human lung cancer cell lines. (A) using method 1, (B) using method 2. (C) Relative *Nanog* transcript in human lung cancer cell lines.

(3) *Is there a possibility that NANOG may be important in mouse lung cancer but not in human lung cancer? If so, it might be good to investigate the mechanism of the difference.*

Response: Thanks for this comment. Our data together with several previous studies showed that NANOG protein was expressed in part of human lung cancer. Our IHC results revealed that NANOG was expressed in around 20% of human ADC and only 4% of human IMA. Importantly, we demonstrated that loss of *Nanog* could induce IMA occurrence in mouse model. The results indicate that NANOG is important both in human and mouse lung cancer.

(4) *Is there any mechanistic study? The data shown by the authors indicate that NANOG inhibits the expression of genes related to gastric differentiation. How does NANOG inhibit these genes? What's the molecular mechanism? The authors may be able to identify the mechanism in vitro.*

Response: Thanks for this very constructive comment. We have indeed tried very hard to look into the mechanisms involved in the regulation of gastric differentiation by NANOG knockout. We used RNAi to

knockdown NANOG expression in human cancer cell lines and checked the expression of various gastric differentiation related genes. Our results showed that knockdown of NANOG had little or no impact upon gastric-related genes expression in two cell lines analyzed (Referee Figure 4). We reason that the *in vitro* 2D culture system might not be suitable to study the mechanism in link to gastric differentiation. Future efforts using organoids culture might be an opportunity to look deep into the related mechanisms.

Referee Figure 4. Relative transcripts of *NANOG*, *HNF4α*, *PDX1*, *LGALS4*, *VSIG1* in human lung cancer cell lines. * $p < 0.05$, **** $p < 0.0001$.

(5) Is *HES1*, *HES5*, *HEY1* or *HEY2* expressed in human IMA? If so, please show the expression level in Figure 3D.

Response: Thanks for this suggestion. We employed *HES1*, *HES5* and *HEY1* antibodies and performed IHC staining in human lung IMA (due to limited number of remaining tissue sections, we analyzed their expression in 20~30 IMA samples) (new Figure EV4). We found that low expression of *HES1* in most of human ADC and IMA (new Figure EV4A). *HEY1* was barely detectable in human ADC and IMA tissues (new Figure EV4A). *HES5* was expressed in around 50% of human ADC and 87% of human IMA (new Figure EV4B). These results indicated an increased expression of *HES5* in human IMA. We further found that *HES5* was expressed in 68.7% of IMA with concurrent loss of *NANOG*, *LKB1* and *NKX2-1* (new Figure EV4C). Since Figure 3 didn't mention the role of Notch pathway, we added these data into Figure EV4.

We also assessed *HES5* expression in KL and KLN mice tumors. *HES5* was highly expressed in KL ADC and KLN IMA tissues (Referee Figure 5A). LY-411575 treatment did not block *HES5* expression (Referee Figure 5B), indicated that *HES5* may play a role in murine lung ADC independent of the Notch pathway.

Referee Figure 5. (A) Representative photos of H&E and HES5 IHC staining in KL ADC, KLN IMA and KLN mice tumors treated with LY-411575. Scale bar, 50 μ m. (B) Statistic analysis of ratio of HES5 positive cells in three groups. * $p < 0.05$.

(6) What's the expression of NKX2-1 in the KLN mouse lungs treated with LY-411575?

Response: We thank reviewer for this question. We evaluated NKX2-1 expression using IHC staining in KLN mouse lung tissues treated with LY-411575 (Referee Figure 6). We found that NKX2-1 was expressed in tumors post LY-411575 treatment.

Referee Figure 6. Representative photos of H&E and NKX2-1 IHC staining in KLN mice tumors treated with LY-411575. Scale bar, 50 μ m.

(7) Did LY-411575 influence airway cilia?

Response: Thanks for this question. It has been reported that Notch signaling established the balance of ciliated and secretory cell fates in airway differentiation (Tsao et al, 2009). Inhibition of Notch pathway

(Jagged antagonism) induced loss of club cells and gain of ciliated cells in lung airway (Lafkas et al, 2015). In our study, we only collected lung lobes when we sacrificed mice. We detected cilia using acetylated-tubulin antibody (Sigma, T6793) in bronchioles (Referee Figure 7). Our results showed that there was no obvious difference of acetylated-tubulin level in vehicle or LY-411575 treated groups. We did not find any report about the effect of LY-411575 in airway cilia. LY-411575 has been reported to improve cochlear hair cell regeneration and induce partial recovery of hearing (Mizutari et al, 2013). The effects of this drug still need more exploration.

Referee Figure 7. Representative photos of IF staining for acetylated-tubulin in lungs from vehicle or LY-411575 treated KLN mice. Scale bar, 50 μ m.

(8) *It would be good to show low magnified images of AB and PAS as well as H&E in Figures 5B and 6A (only H&E image is shown now), which would help the readers to get an idea about the degree of prevalence of IMA in the mouse lungs.*

Response: Thanks for this suggestion. We have changed the figures (new Figure 5F and 6A).

(9) *Winslow et al (2011) should be cited in this part of the Introduction: The transcription factor NKX2-1 generally expresses in human lung adenocarcinoma (ADC) but decreases with poor differentiation and malignant progression (Boland et al, 2018; Cai et al, 2014; Cha & Shim, 2017).*

Response: Thanks for this comment. We have cited this paper (Line 55).

(10) *Bar graphs with dot plots are recommended to ensure data integrity.*

Response: Thanks for this suggestion. We have changed the type of graphs (new Figures 2-6).

Referee #2 (Comments on Novelty/Model System for Author):

1. RNA-seq data showed that stem cell pluripotency expression pattern was significantly enriched in KL tumors in contrast to K or KP tumors (Fig 1A and B). These data are expectable because KL tumor exhibits strong plasticity and lineage transition from adenocarcinoma (ADC) to squamous cell carcinoma (SCC) in previous literatures.

2.Generation of Nanog conditional knockout mice which show mucinous differentiation is a great work!

3.Combination therapy with phenformin and Notch inhibitor LY-411575 has anti-tumor effect in KLN mice through inhibiting mucinous lineage transition. However, clinical application of this combination therapy in human IMA, which cancer cells already existed, is doubtful.

Referee #2 (Remarks for Author):

This manuscript by Xinyuan Tong et al. used Kras-driven lung cancer genetically engineered mouse model (GEMM) to investigate an undruggable subtype, invasive mucinous adenocarcinoma (IMA). They demonstrated that Nanog deletion in LKB1 deficiency lung adenocarcinoma mouse model promoted mucinous differentiation of tumor, which could be blocked by Notch inhibitor, LY-411575. They also showed the frequently concurrent loss of NANOG and LKB1 expression in human IMA. Finally, they displayed that IMA was more resistant to phenformin treatment, an analog of anti-diabetes drug metformin, compared to that in adenocarcinoma. This resistance could be overcome by LY-411575.

The merits of this study are GEMM and abundant human lung cancer tissue. However, I still have some concerns about experiments.

Major comments:

1. According to the result of "Nanog deletion in KL tumors triggers gastric differentiation", the expression of LKB1 and Nanog should be investigated in human adenocarcinoma and IMA. Only Nanog (Fig.3C) in human lung cancer tissue is not enough.

Response: We thank the reviewer for this comment. We have added LKB1 IHC staining and performed statistical analysis (new Figure 3D-F). The results showed that LKB1 was expressed in around 24% of human ADC and in 9% of human IMA. The detailed description has been added into results (Lines 208-217).

2. To investigate the different resistance of mucinous tumor and adenocarcinoma to phenformin (which can selectively target LKB1-deficient ADC), phenformin should be used in KL and KLN mice. All lung tumors in KL mice supposed to be LKB1-deficient ADC and all tumors in KLN mice supposed to be LKB1 and Nanog deficient. I wondered the experimental design in Fig5. Moreover, author state "little effect was observed on the occurrence of mucinous tumors after phenformin treatment (Fig 5B and C)", but the result in Fig.5C seems not! The mice number of Fig.5C (5 vs. 7, vehicle vs. phenformin) should be equal.

Response: We thank the reviewer for this comment. We have now treated KL mice with phenformin for 2 weeks and did the statistics analysis. The results showed that phenformin blocked cell proliferation of ADC in KL mouse model (new Figure 5A-C), consistent with our previous work (Li et al, 2015).

Regarding the effects of phenformin treatment in KLN mice, we have compared the IMA numbers in vehicle group and phenformin treatment group. The results indicated that IMA numbers had no significant change (Referee Figure 8). We have also re-performed this experiment and combined these two independent results (new Fig 5F).

Referee Figure 8. Statistics analysis of IMA number (no.) in KLN mice treated with or without phenformin. Significance was calculated by unpaired Student's t-test.

Referee #3 (Remarks for Author):

The study by Tong et al titled *Nanog maintains stemness of Lkb1-deficient lung adenocarcinoma and prevents gastric differentiation utilizes GEMM models to further characterize the genetic components involved in invasive mucinous adenocarcinoma development*. This manuscript is well written and provides a number of novel findings including the role of Nanog.

I do however have some concerns and these are listed below • I know the numbers aren't statistically significant but it appears as if KLN mice have more small tumors (increased tumor number but decreased tumor burden). I'm not sure if this is true or the significance of this observation if it is true but the authors might want to evaluate this further.

Response: We thank the reviewer for this comment. We classified the tumors into different grades according to histological features (Tammela et al, 2017) (Figure EV2D). Also, we distinguished the tumors into small tumors and large tumors (Figure EV2E). The results indicated that there were more adenomas (grade 1 and grade 2) and more small tumors ($<0.1\text{mm}^2$) in KLN mice. We found no significant change in tumorigenesis when we compared KLN group with KL group.

According to the pictures of whole lung lobes of these mice (Referee Figure 9) and statistical analysis of tumor grade, tumor burden and tumor subtypes, there was no significant change of tumorigenesis in KLN mice compared with KL mice. The description has been added into the result part (Lines 151-153).

Referee Figure 9

Referee Figure 9. Photos of lung lobes of KL and KLN mice (n=6 per group). Scale bar, 0.5cm.

• Some more information about the IMA tumors would be appreciated. Are they actually invasive? Do they metastasize? A more detailed description of the tumor features, as described by a pathologist, should be provided.

Response: We thank the reviewer for this comment. Human IMA contained amount of invasion (>0.5 cm) (Travis et al, 2011). Human and murine IMA were loss of a circumscribed border and slightly spread into adjacent lung parenchyma. Among the IMAs with invasive lesions, papillary predominant patterns and acinar predominant patterns were observed in our human and murine tissues. These tumors pattern of mouse type were confirmed by the professional pathologist in the department of thoracic surgery.

IMA also metastasized. Previous study showed that extrathoracic versus lung metastasis did not distinguish between the mucinous and non-mucinous subtypes of adenocarcinoma (Wislez *et al*, 2010). We did not observe significant difference in metastasis between KL and KLN mouse model. As suggested, we have provided a more detailed description of the features of the IMA tumors in our revised manuscript (Lines 161-168).

• I'm not sure I would agree with the statement on page 8 that "We found that the mucinous ADC closely recapitulated the histological characteristics of human lung IMA". From the information presented, there are some similarities but also a lot of structural differences. I think additional measures or description of the shared histological features are required to support this claim.

Response: Thanks for this comment. As suggested, we have added representative H&E photos of human and murine IMA in new Figure 2A and B. They were both composed of columnar or goblet cell morphology with abundant mucus in cytoplasm and showed small basal oriented nuclei. These tumors showed the same heterogeneous mixture of acinar or papillary growth patterns as in non-mucinous tumors. Also, these tumor patterns of mouse type were confirmed by the professional pathologist in the department of thoracic surgery. We have added the detailed description in results (Lines 161-168).

• For the qRT-PCR analyses of the tumors (ie Fig 2F), how was the RNA extracted? Was laser capture microdissection or some sort of sorting system used? If bulk lungs were used, how did the authors select the lungs to ensure parameters like similar tumor burden, enrichment for ADC or IMA in the samples used, etc.

Response: Thanks for this comment. We used freshly dissected tumors for the qRT-PCR analyses. We collected lots of tumors from KL and KLN mouse model at 8 weeks post Ad-Cre treatment for RNA extraction. Using these RNAs, we then performed real-time PCR to evaluate the expression of *Muc5ac*, a mucin-related marker gene (Copin et al, 2001), which helped us to distinguish the tumors with or without mucin. We have now clarified this in the method part (Lines 433-435).

• *Only 5 of the 9 gastric differentiation genes were upregulated. So, what does this mean when almost half of the gastric differentiation genes were not significantly upregulated? Also, why did the applicants use GKN1 in the immunohistochemistry when this marker was not upregulated at the mRNA level?*

Response: Thanks for this comment. Indeed, we observed that not all gastric differentiation genes were up-regulated as mentioned by the reviewer. We reason that this could be potentially due to the complicated regulation during *in vivo* tumor differentiation. Of note, *Hnf4a*, the master regulator for gastric lineage switch (Camolotto et al, 2018; Snyder et al, 2013), was up-regulated during gastric differentiation. Besides, GKN1 is reported to be a common marker gene in stomach and highly expressed in mucinous tumors (Snyder et al, 2013). Our previous work also supports an important role for GKN1 in lung cancer (Yao et al, 2019). Although mRNA level of GKN1 was not up-regulated, we still checked its protein level according to previous literature (Snyder et al, 2013). Interestingly, we found that GKN1 protein was up-regulated in mucinous tumors. The inconsistency between GKN1 mRNA and protein levels might be explained by the post-transcriptional level regulation which requires further study to clarify.

• *Unfortunately, statistics is not an area of expertise for me however, I think the statistics in this paper should be evaluated more closely. For example, a Student's t-test is not a suitable statistical test for comparing means of 3 groups such as the data in Fig 2C, Fig 5E,F, Fig 6F,G. Also given the large error bars in Fig 4C,D I find it improbable that these means are statistically different unless a very large number of samples were used. More details on how these counts were performed and analyzed, should be provided.*

Response: Thanks a lot for this valuable comment. As suggested, we have now re-evaluated the statistics of our results using appropriate statistical methods. For comparing means of 2 groups, significance was determined using two-tailed unpaired Student's t-test with Welch's correction. For comparing means of 3 groups, significance was determined using one-way ANOVA with Dunnett's multiple comparisons test. For comparing the grade of IHC staining, significance was determined using χ^2 test. We have corrected our statistic results whenever possible in our manuscript and added details into legends and methods about the statistical methods used (Lines 459-464).

Regarding how counts were performed and analyzed, we have changed the type of graphs to present the data using bar graphs with dot plots. In new Figure 4C and D, each dot plot represented a ratio of NICD or HES1 positive cells to all tumor cells in an individual tumor. Around 15~18 tumors were taken pictures and

calculated in each group. We used two-tailed unpaired Student's t-test with Welch's correction to compare these two groups. We have also added these details into method (Lines 408-410, 459-464).

Once again, thanks a lot for all these valuable comments and suggestions.

28th Aug 2020

Dear Prof. Ji,

Thank you for submitting your revised manuscript to EMBO Molecular Medicine. We have now heard back from the three referees who agreed to evaluate the new version of your manuscript.

As you will see from the reports below, while referees #2 and #3 are satisfied with the revisions and are now supportive of publication, referee #1 is still not convinced that NANOG is expressed in human lung cancer. As mentioned in my previous decision letter, human relevance is a prerequisite for publication in EMBO Molecular Medicine. EMBO Press encourages a single round of revisions only, and we would therefore normally reject the manuscript at this stage. However, should you be able to convincingly demonstrate NANOG expression in human lung cancer tissue, we would welcome resubmission of your revised manuscript.

Specifically, please address points the different points from referee #1:

- 1/ provide qualitative and quantitative data of NANOG expression in healthy and cancer human lung tissue, as well as mouse cancer tissue.
- 2/ provide unambiguous western blot data (several independent experiments) and confirm antibody specificity.
- 3/ clarify NANOG expression data in the different cell lines in regard to public expression data.

To be entirely clear though, should you not be able to address these points and demonstrate the relevance of your data for human disease, we will not be able to offer further consideration to your manuscript in EMBO Molecular Medicine. Should you find that the requested revisions are not feasible and prefer, therefore, to submit your paper elsewhere, we would welcome a message to this effect.

When submitting your revised manuscript, please carefully review the instructions that follow below. Failure to include requested items will delay the evaluation of your revision:

- 1) A .docx formatted version of the manuscript text (including legends for main figures, EV figures and tables). Please make sure that the changes are highlighted to be clearly visible.
- 2) Individual production quality figure files as .eps, .tif, .jpg (one file per figure).
- 3) A .docx formatted letter INCLUDING the reviewers' reports and your detailed point-by-point responses to their comments. As part of the EMBO Press transparent editorial process, the point-by-point response is part of the Review Process File (RPF), which will be published alongside your paper.
- 4) Please note that all corresponding authors are required to supply an ORCID ID for their name

upon submission of a revised manuscript.

5) We would also encourage you to include the source data for figure panels that show essential data. Numerical data should be provided as individual .xls or .csv files (including a tab describing the data). For blots or microscopy, uncropped images should be submitted (using a zip archive if multiple images need to be supplied for one panel). Additional information on source data and instruction on how to label the files are available at

6) Please upload EV figures separately. There is a file with EV figures and legends and an EV table. The EV figure legends need to be added to the main manuscript, after the main figure legends. EV figures should be uploaded separately as individual, high-resolution figure files. The EV table should be uploaded as a separate file, in word or excel format, with its legend added directly in the file.

7) The paper explained: EMBO Molecular Medicine articles are accompanied by a summary of the articles to emphasize the major findings in the paper and their medical implications for the non-specialist reader. Please provide a draft summary of your article highlighting

8) For more information: There is space at the end of each article to list relevant web links for further consultation by our readers. Could you identify some relevant ones and provide such information as well? Some examples are patient associations, relevant databases, OMIM/proteins/genes links, author's websites, etc...

9) Every published paper now includes a 'Synopsis' to further enhance discoverability. Synopses are displayed on the journal webpage and are freely accessible to all readers. They include a short stand first (maximum of 300 characters, including space) as well as 2-5 one-sentences bullet points that summarizes the paper. Please write the bullet points to summarize the key NEW findings. They should be designed to be complementary to the abstract - i.e. not repeat the same text. We encourage inclusion of key acronyms and quantitative information (maximum of 30 words / bullet point). Please use the passive voice. Please attach these in a separate file or send them by email, we will incorporate them accordingly.

Please also suggest a striking image or visual abstract to illustrate your article. If you do please provide a png file 550 px-wide x 400-px high.

10) As part of the EMBO Publications transparent editorial process initiative (see our Editorial at <http://embomolmed.embopress.org/content/2/9/329>), EMBO Molecular Medicine will publish online a Review Process File (RPF) to accompany accepted manuscripts.

In the event of acceptance, this file will be published in conjunction with your paper and will include the anonymous referee reports, your point-by-point response and all pertinent correspondence relating to the manuscript. Let us know whether you agree with the publication of the RPF and as here, if you want to remove or not any figures from it prior to publication.

I look forward to receiving your revised manuscript.

Yours sincerely,

Lise Roth
Lise Roth, PhD
Editor
EMBO Molecular Medicine

To submit your manuscript, please follow this link:

Link Not Available

***** Reviewer's comments *****

Referee #1 (Remarks for Author):

Comments

Unfortunately, I am still not convinced by the data that NANOG is expressed in human lung cancer.

(1) Please compare Figure 1D (bottom panel) with Figure 3B (bottom panel). NANOG is highly expressed in the nucleus of KL mouse lung cancer cells in Figure 1D but I do not see such expression in the human lung cancer cells (ADC) in Figure 3B.

(2) In Referee Figure 2, the authors showed their western data done by using extracts of mouse embryo stem cells (ESC), normal lung (mouse or human?) and human lung tumor (what kind of lung tumor: adeno, squamous, small?). First, this is obviously an n=1 experiment. The authors need more n to prove their point. Second, using the Bethyl antibody, which is supposed to detect mouse NANOG, the expression of NANOG in normal lung is comparable to that in mouse ES cells. And the expression of NANOG in human lung tumor is much higher than that in mouse ES cells. Is this possible considering NANOG is an ESC transcriptional factor? The CST antibody is supposed to detect human NANOG, but it is detecting mouse NANOG in mouse ES cells. Ji's lab is good, so they could easily do simple gain-of-function experiments using a mammalian expression vector that carries mouse or human cDNA of NANOG and test these antibodies with or without the expression vector to confirm the antibody specificity instead of doing n=1 human specimen experiment.

Out of curiosity, I looked up other ESC transcription factors (OCT4 and SOX2). OCT4 (POU5F1) is actually similar to NANOG, and both of them are highly expressed in testis cancer (<https://www.proteinatlas.org/ENSG00000204531-POU5F1/pathology>). SOX2 is highly expressed in lung cancer (<https://www.proteinatlas.org/ENSG00000181449-SOX2/pathology>) and, to my knowledge, there are lots of reports on SOX2 in the lung cancer field.

(3) Regarding the cell line experiment, I looked up EMBL-EBI Expression Atlas (<https://www.ebi.ac.uk/gxa/home>). The authors claim NANOG is expressed in H358, H1299 and H2126 human lung cancer cell lines. However, according to the Expression Atlas, the expression of NANOG in H358, H1299 and H2126 is below cutoff (I couldn't find the expression of NANOG in H1975 by my quick search). Are the authors indeed confident about their claim that NANOG is expressed in human lung cancer? TCGA dataset used in the Human Protein Atlas does not support the authors' claim either (<https://www.proteinatlas.org/ENSG00000111704-NANOG/pathology>), which even states "enriched in testis cancer".

(4) I think that the mouse data by the authors seems convincing. NANOG may be involved in mouse lung cancer pathogenesis while it is not in human lung cancer pathogenesis. There are differences in lung pathogenesis between human and mouse. For example, human patients who carry defective CFTR develop diseases in intestine and lung but CFTR-knockout mice develop disease in intestine but not in lung (<https://www.ncbi.nlm.nih.gov/pmc/articles/PMC508519/>). Hence, the data by the authors is useful for researchers in case they happen to develop mice with lung tumors with or without gastric differentiation then they can look at the status of NANOG in addition to other gastric differentiation markers.

Referee #2 (Comments on Novelty/Model System for Author):

Authors made great efforts to answer reviewers' questions, including evaluation of public RNA-Seq dataset from Chinese lung ADC, re-perform IHC staining with new antibodies, repeat animal studies, and modifying statistical methods. Revised manuscript seems more logically and convincingly.

Referee #2 (Remarks for Author):

Authors responded these concerns specifically and added many convincing data in revised manuscript. I have no questions.

Referee #3 (Remarks for Author):

My concerns have been addressed by the authors

Re: EMM-2020-12627-V2

Nanog maintains stemness of *Lkb1*-deficient lung adenocarcinoma and prevents gastric differentiation

***** *Reviewer's comments* *****

Referee #1 (Remarks for Author):

Comments

Unfortunately, I am still not convinced by the data that NANOG is expressed in human lung cancer.

Response: We appreciate the Referee's comment! To address this concern, we have now performed a serial of new experiments including qRT-PCR for *NANOG* in human and mouse normal lung and lung tumors, RT-PCR for *NANOG* in human cancer cell lines and human ADC samples to distinguish *NANOG* and its pseudogenes, western blot in murine and human cancer cell lines with *NANOG* overexpression to confirm antibody specificity and western blot for endogenous *NANOG* in human lung ADC specimens and paired pathologically normal lung tissues. Below we will list our data.

Firstly, we performed qRT-PCR to detect *NANOG* transcripts in 15 human lung ADC samples and paired pathologically normal lung tissues. We found that about 26.7% (4/15) of lung tumors had the *NANOG* transcriptional expression at about 10 fold lower than that detected in human embryonic stem cells (ESCs) (Referee Figure 10A). There were 40% (6/15) of lung ADCs expressed approximately >500 fold less *NANOG* transcripts compared with human ESCs (Referee Figure 10A). To our surprise, around half of human normal lungs showed *NANOG* expression comparable to paired human ADC (Referee Figure 10A). This unexpected observation promoted us to further check public TCGA database (<http://ualcan.path.uab.edu/index.html>). In TCGA dataset, we found that compared to human lung ADCs, normal lungs did show similar levels of *NANOG* transcriptional expression (Referee Figure 10B). Although we couldn't

provide a good explanation for these observations, these results do support the presence of *NANOG* transcripts in human normal lung tissues and ADC samples albeit at magnitudes lower than human ESCs. Future efforts will be interesting to look more details into these findings.

Referee Figure 10. (A) Relative *NANOG* transcripts in 15 human lung ADC samples and paired normal lung tissues compared to hESCs. Red line indicated 1/10 of *NANOG* transcripts of hESCs. (B) Transcript per million of *NANOG* in human normal lung and primary ADC tissues from TCGA database (<http://ualcan.path.uab.edu/cgi-bin/TCGAExResultNew2.pl?genenam=NANOG&ctype=LUAD>).

Human *NANOG* gene belongs to a gene family containing a gene tandem duplication (named *NANOG2* or *NANOGP1*) and several pseudogenes (*NANOGP2-P11*), among which *NANOG* and *NANOGP8* are considered to be translated into proteins (Eberle et al, 2010; Palla et al, 2014). *NANOGP8* shares 99.5% homology to *NANOG* open reading frame (ORF) and codes for a 305-amino acid protein which differs from *NANOG* by only 3 amino acids (Palla et al, 2014) (Referee Figure 11A). Ambady *et al.* have previously demonstrated that Primer 1 mainly amplified *NANOG* and *NANOGP8* (387bp), together with a lower band designating *NANOGP5* with the 38bp deletion. Using *NANOG*-387 primers (Primer 1) from Ambady's study, we performed RT-PCR and detected the expected 387bp fragments in human normal lungs and paired lung tumors (Referee Figure 11B). This 387bp fragment was also detected in human ESCs (Referee Figure 11B). We confirmed the PCR products from human lung ADCs was from *NANOG* ORF region using Sanger Sequencing. We also observed the lower band

as *NANOGP5* (Referee Figure 11B), consistent with previous report.

Ambady *et al.* have developed a method to distinguish the *NANOG* and its pseudogene *NANOGP8* using PCR with the NANOG-1860 primers (Primer 2) followed by *Sma* I restriction enzyme digestion (Ambady *et al.*, 2010). Although the 1860bp fragments can be amplified from both *NANOG* and *NANOGP8* transcripts, only the PCR product from *NANOG* transcript can produce two fragments of 1236bp and 624bp after *Sma* I digestion (Ambady *et al.*, 2010; Park *et al.*, 2019). In contrast, the PCR product from *NANOGP8* transcript will remain as 1860bp after *Sma* I digestion due to the absence of *Sma* I cutting site (Referee Figure 11C). Our data showed that only two fragments (1236bp and 624bp bands) were detected in human lung ADC, normal lung tissues and human ESCs after *Sma* I digestion, indicating these lung samples contained only *NANOG* transcript but no *NANOGP8* transcript, similar to human ESCs. In contrast, all five human lung cancer cell lines showed three fragments (1860bp, 1236bp and 624bp band), indicating that they contained both *NANOG* and *NANOGP8* transcripts (Referee Figure 11D). Although we don't know how this happens, these results together demonstrate that the *NANOG* transcripts in human lung ADC and normal lung tissues are mainly derived from *NANOG* instead of its pseudogene *NANOGP8*.

Referee Figure 11

Referee Figure 11. (A) Schematic illustration of partial regions identified by NANOG-387 primers (Primer 1) and NANOG-1860 primers (Primer 2) of *NANOG* or its pseudogenes. The pseudogene *NANOGP5* has many base changes (not indicated). (B) RT-PCR for *NANOG* transcripts using Primer 1 from 15 paired human normal lung and lung ADC tissues. *GAPDH* was used as the control. N, normal lung. T, lung ADC tumor. (C) Alignment of 3' UTR regions (partial sequence) of *NANOG* and *NANOGP8*. The restriction site for Sma I (CCCGGG) in *NANOG* is absent in *NANOGP8* due to a C to G transversion (shaded region). (D) RT-PCR for *NANOG* using Primer 2 from human cancer cell lines and human normal lungs and ADC samples. Sma I digestion of the 1860bp RT-PCR product from *NANOG* transcript produced 1236bp and 624bp fragments whereas Sma I digestion of the RT-PCR product from *NANOGP8* transcript produced only 1860bp fragment. N, normal lung. T, ADC tumor.

To confirm the specificity of mouse antibody (the Bethyl antibody), we overexpressed full-length mouse NANOG cDNA in mouse cell lines. Using the Bethyl antibody, we found that the exogenous NANOG expression produced 4 different bands, two around 40kDa and another two around 35kDa (Referee Figure 12A). We reason that the appearance of multiple bands might be due to different post-translational modifications of NANOG. In comparison with Kras mefs and KP cells, the KL cells showed a 40kDa upper band at the same size as that observed in mouse ESC, indicative of endogenous NANOG expression. This 40kDa upper band was also detectable in the exogenous NANOG overexpression groups (Referee Figure 12A). These data suggested that the Bethyl antibody could recognize both endogenous and exogenous mouse NANOG. In contrast to Kras mefs and KP cells, the KL cells showed relatively low level of exogenous NANOG expression. We reason that the endogenous level of NANOG might be sufficient to maintain KL cancer stemness and somehow suppresses the exogenous NANOG expression via unknown feedback loop.

To further test the specificity of mouse antibody (the Bethyl antibody), we also overexpressed NANOG with fused 3xFLAG tag at C-terminal. Western blot results using the NANOG antibody (Bethyl) and FLAG antibody (Sigma, F3165) showed similar expression pattern for NANOG in these cell lines (adding 3xFLAG tag is about to add 9kDa) (Referee Figure 12B). These results further confirm that the Bethyl

antibody could specifically detect the NANOG expression.

To test the specificity of human antibody (the CST antibody), we overexpressed full-length human NANOG cDNA in human lung cancer cell lines. Using the CST antibody, we found that the exogenous NANOG expression produced 3 different bands, two around 40kDa and another one around 25kDa (Referee Figure 12C). Western blot of one cell line H358 showed the 40kDa lower band, indicative of endogenous NANOG expression. This 40kDa lower band was also observed in NANOG overexpression groups (Referee Figure 12C). We further overexpressed NANOG with fused HA tag at N-terminal. Western blot data using the NANOG antibody (CST) and HA antibody (Absci, AB35534) showed similar expression pattern (adding HA tag is about to add 3kDa) (Referee Figure 12D). Interestingly, we noticed that there existed different expression patterns of exogenous NANOG in three different cell lines. For example, only one 40kDa upper band was observed in A549 cells (also detected in human ESC) whereas the other two cell lines showed three different bands. We speculate that the post-translation modification of NANOG is quite complicated and might change with different cell lines.

Taken together, these new data indicate that the Bethyl antibody and the CST antibody can specifically recognize endogenous mouse NANOG and human NANOG respectively. The post-translational modifications of NANOG seem quite complicated and future efforts will be necessary to clarify the underlying mechanisms.

Referee Figure 12. (A) Western blot for NANOG and TUBULIN in mouse cell lines (*Kras*^{G12D/+} Mefs, *Kras*^{G12D/+}; *P53*^{-/-} (KP) primary tumor cell lines, *Kras*^{G12D/+}; *Lkb1*^{-/-} (KL) primary tumor cell lines) transfected with empty vector or mouse NANOG overexpression vector. The Dot indicated the non-specific band. The arrow indicated the band of exogenous mouse NANOG. The asterisk indicated the band of endogenous mouse NANOG. (B) Western blot for FLAG and TUBULIN in mouse cell lines transfected with empty vector or overexpression vector with NANOG fused with 3xFLAG tag in C-terminal. The Dot indicated the non-specific band. The arrow indicated the band of exogenous FLAG. (C) Western blot for NANOG and TUBULIN in human adenocarcinoma cell lines transfected with empty vector or human NANOG overexpression vector. The arrow indicated the band of exogenous human NANOG. The asterisk indicated the band of endogenous human NANOG. (D) Western blot for HA and TUBULIN in human cancer cell lines transfected with empty vector or overexpression vector with NANOG fused with HA tag in N-terminal. The Dot indicated the non-specific band. The arrow indicated the band of exogenous HA.

Moreover, we have used the CST antibody to detect endogenous NANOG expression in human lung ADC samples and paired normal lungs. According to our previous IHC staining data, these human samples were found to have medium or high expression of

NANOG in tumors. Consistently, we observed that NANOG was expressed in most of these human lung ADCs at protein level, but not in paired normal lungs (Referee Figure 13). These results were different from the NANOG expression at RNA level in human lung tissues, indicating that post-transcriptional modifications of NANOG might affect NANOG protein abundance. NANOG is an important transcription factor in ESCs. It can also be used as a marker to evaluate cancer stemness (Chang et al, 2017; Chiou et al, 2010; Koh et al, 2019; Maiuthed et al, 2018; Noh et al, 2012). We observed that the location of the band of endogenous NANOG in human lung tissues was a little lower than the band in hESCs. This might be due to the different post-translational modifications of NANOG and future efforts will be necessary to clarify more into details.

Referee Figure 13. Western blot for NANOG and TUBULIN in paired human normal lungs and ADC samples and human ESCs.

Overall, NANOG is expressed in human lung cancer but more experiments and suitable systems are needed to clarify the post-transcriptional and post-translational modifications of NANOG in human lungs. Our new data here have provided several clues: (1) Expression of NANOG at protein level might be different from RNA level in human lung cancer and normal lung tissues; (2) Human lung cancer and normal lung tissues mainly express the *NANOG* transcript whereas human lung cancer cell lines express both *NANOG* and *NANOGP8* transcripts; (3) Post-transcriptional and post-translational modifications of NANOG in normal cells and tumor cells are quite complicated.

(1) Please compare Figure 1D (bottom panel) with Figure 3B (bottom panel). NANOG is highly expressed in the nucleus of KL mouse lung cancer cells in Figure 1D but I do not see such expression in the human lung cancer cells (ADC) in Figure 3B.

Response: Thanks for pointing out this difference between human and mouse cancer cells. Based on our results (Referee Figure 12), we found the Bethyl antibody and the CST antibody could detect endogenous NANOG specifically. The different sub-cellular localization of NANOG might be influenced by different post-translational modifications of NANOG. The localization of NANOG in tumor cells might also be affected by differentiation grade of tumors (Li et al, 2013a; Li et al, 2013b).

(2) In Referee Figure 2, the authors showed their western data done by using extracts of mouse embryo stem cells (ESC), normal lung (mouse or human?) and human lung tumor (what kind of lung tumor: adeno, squamous, small?). First, this is obviously an n=1 experiment. The authors need more n to prove their point. Second, using the Bethyl antibody, which is supposed to detect mouse NANOG, the expression of NANOG in normal lung is comparable to that in mouse ES cells. And the expression of NANOG in human lung tumor is much higher than that in mouse ES cells. Is this possible considering NANOG is an ESC transcriptional factor? The CST antibody is supposed to detect human NANOG, but it is detecting mouse NANOG in mouse ES cells. Ji's lab is good, so they could easily do simple gain-of-function experiments using a mammalian expression vector that carries mouse or human cDNA of NANOG and test these antibodies with or without the expression vector to confirm the antibody specificity instead of doing n=1 human specimen experiment.

Out of curiosity, I looked up other ESC transcription factors (OCT4 and SOX2). OCT4 (POU5F1) is actually similar to NANOG, and both of them are highly expressed in testis cancer (<https://www.proteinatlas.org/ENSG00000204531-POU5F1/pathology>). SOX2 is highly expressed in lung cancer (<https://www.proteinatlas.org/ENSG00000181449-SOX2/pathology>) and, to my knowledge, there are lots of reports on SOX2 in the lung cancer field.

Response: Thank you for this question. In previous Referee Figure 2, we used extracts of mouse ESCs, one paired human normal lung and ADC tumor (N15 and T15 used in

Referee Figure 11). We compared the Bethyl antibody and the CST antibody and chose the CST antibody to do further experiments because the CST antibody showed better performance in recognizing human NANOG. In the CST antibody study, we did not observe NANOG expression in human normal lungs. It's our mistake to use mouse ESCs as control because the protein abundance of NANOG in mESCs and hESCs might be different. We don't know why the CST antibody could recognize mouse NANOG but we found that there were two papers using this antibody to detect mouse NANOG (<https://www.labome.com/product/Cell-Signaling-Technology/4903.html>). To address the Referee's concern, we re-performed western blot to detect NANOG in human cell lines transfected with overexpression vectors to confirm antibody specificity and collected more paired human normal lungs and lung ADC to check NANOG expression (Referee Figure 12 and 13).

(3) Regarding the cell line experiment, I looked up EMBL-EBI Expression Atlas (<https://www.ebi.ac.uk/gxa/home>). The authors claim NANOG is expressed in H358, H1299 and H2126 human lung cancer cell lines. However, according to the Expression Atlas, the expression of NANOG in H358, H1299 and H2126 is below cutoff (I couldn't find the expression of NANOG in H1975 by my quick search). Are the authors indeed confident about their claim that NANOG is expressed in human lung cancer? TCGA dataset used in the Human Protein Atlas does not support the authors' claim either (<https://www.proteinatlas.org/ENSG00000111704-NANOG/pathology>), which even states "enriched in testis cancer".

Response: Thanks Referee for this comment. We checked the public database in EMBL-EBI Expression Atlas. We downloaded the TPM (Transcripts Per Million reads) results of *NANOG* in 146 human lung cancer cell lines. Also, we found the public dataset of *NANOG* expression in 254 human lung cancer cell lines from cBioPortal (<https://www.cbioportal.org>). The expression of *NANOG* in several lung adenocarcinoma cell lines was low but could still be detected (Referee Figure 14). We confirmed that part of cell lines including H358, H2126, H1944 and H1975 cells express *NANOG* at RNA level. We reason that the difference between different databases might be due to the sequencing depth and coverage.

We also checked the protein abundance of NANOG in Human Protein Atlas as well as the antibody they used (<https://www.proteinatlas.org/ENSG00000111704-NANOG/antibody>). However, we couldn't find the information of this antibody (CAB019380). We tried to get the detailed product information from the provider (Origene). There were several NANOG antibodies from the Origene website. However, we could find only 1 citation from their products (Lin et al, 2019). It seems difficult for us to confirm if the antibody works well or not.

Nonetheless, our new data reveal that NANOG is expressed in human lung tissues and cancer cell lines at RNA level and protein level. We speculate that NANOG is not a high-abundance transcript but biologically important and the post-transcriptional and post-translational modifications of NANOG in lung cancer need more future investigations.

Referee Figure 14

Sample Name	NANOG(RNA-seq TPM)
lung, non-small cell lung carcinoma, NCI-H1155	0.5
lung, non-small cell lung carcinoma, NCI-H1770	0.5
lung, lung adenocarcinoma, HCC364	0.3
lung, lung adenocarcinoma, HCC515	0.3
lung, non-small cell lung carcinoma, LXFL529	0.3
lung, non-small cell lung carcinoma, NCI-H1703	0.3
lung, non-small cell lung carcinoma, NCI-H1869	0.3
lung, non-small cell lung carcinoma, NCI-H1915	0.3
lung, lung carcinoma, A549	0.2
lung, lung adenocarcinoma, NCI-H1975	0.2
lung, non-small cell lung carcinoma, NCI-H1299	0.2
lung, non-small cell lung carcinoma, NCI-H1437	0.2
lung, non-small cell lung carcinoma, NCI-H2122	0.2
lung, non-small cell lung carcinoma, NCI-H2126	0.2
lung, lung adenocarcinoma, ABC-1	0.2
lung, lung adenocarcinoma, HCC1534	0.2
lung, lung adenocarcinoma, HCC2270	0.2
lung, lung adenocarcinoma, HCC2279	0.2
lung, lung adenocarcinoma, HCC2935	0.2
lung, lung adenocarcinoma, HCC461	0.2
lung, lung adenocarcinoma, LXF-289	0.2
lung, lung adenocarcinoma, NCI-H1395	0.2
lung, lung adenocarcinoma, NCI-H1648	0.2
lung, lung adenocarcinoma, NCI-H1781	0.2
lung, lung adenocarcinoma, NCI-H2009	0.2
lung, lung adenocarcinoma, NCI-H2073	0.2
lung, lung adenocarcinoma, NCI-H820	0.2
lung, lung adenocarcinoma, RERF-LC-KJ	0.2
lung, non-small cell lung carcinoma, NCI-H1944	0.1
lung, non-small cell lung carcinoma, NCI-H358	0.1

(EMBL-EBI)

STUDY ID	SAMPLE ID	NANOG(RNA-seq RPKM)
ccle_broad_2019	NCIH1573 LUNG	0.15839
ccle_broad_2019	HCC1171 LUNG	0.06857
ccle_broad_2019	NCIH358 LUNG	0.06622
ccle_broad_2019	RERFLCAI LUNG	0.05776
ccle_broad_2019	NCIH1666 LUNG	0.05609
ccle_broad_2019	NCIH2122 LUNG	0.05276
ccle_broad_2019	NCIH2405 LUNG	0.05037
ccle_broad_2019	SW1573 LUNG	0.0426
ccle_broad_2019	NCIH1048 LUNG	0.04195
ccle_broad_2019	NCIH889 LUNG	0.04145
ccle_broad_2019	HCC1195 LUNG	0.04063
ccle_broad_2019	SBC5 LUNG	0.04036
ccle_broad_2019	HCC78 LUNG	0.03806
ccle_broad_2019	NCIH2126 LUNG	0.03786
ccle_broad_2019	NCIH2081 LUNG	0.03651
ccle_broad_2019	LCLC97TM1 LUNG	0.03535
ccle_broad_2019	MERO84 LUNG	0.0323
ccle_broad_2019	NCIH1341 LUNG	0.0322
ccle_broad_2019	HCC515 LUNG	0.03168
ccle_broad_2019	NCIH1650 LUNG	0.03083
ccle_broad_2019	NCIH1838 LUNG	0.03015
ccle_broad_2019	NCIH1944 LUNG	0.02819
ccle_broad_2019	NCIH1975 LUNG	0.01604
ccle_broad_2019	NCIH23 LUNG	0.0107
ccle_broad_2019	NCIH1437 LUNG	0
ccle_broad_2019	A549 LUNG	0
ccle_broad_2019	NCIH1299 LUNG	0

(cBioPortal)

Referee Figure 14. (A) TPM of NANOG in several human lung adenocarcinoma cell lines from EMBL-EBL. (B) RPKM (Reads Per Kilobase per Million mapped reads) of NANOG in lung non-small cell lung cancer cell lines from cBioPortal. The highlighted parts were the cell lines we used to detect NANOG expression.

(4) I think that the mouse data by the authors seems convincing. NANOG may be involved in mouse lung cancer pathogenesis while it is not in human lung cancer pathogenesis. There are differences in lung pathogenesis between human and mouse.

For example, human patients who carry defective CFTR develop diseases in intestine and lung but CFTR-knockout mice develop disease in intestine but not in lung (<https://www.ncbi.nlm.nih.gov/pmc/articles/PMC508519/>). Hence, the data by the authors is useful for researchers in case they happen to develop mice with lung tumors with or without gastric differentiation then they can look at the status of NANOG in addition to other gastric differentiation markers.

Response: Thank you for this comment.

Referee #2 (Comments on Novelty/Model System for Author):

Authors made great efforts to answer reviewers' questions, including evaluation of public RNA-Seq dataset from Chinese lung ADC, re-perform IHC staining with new antibodies, repeat animal studies, and modifying statistical methods. Revised manuscript seems more logically and convincingly.

Referee #2 (Remarks for Author):

Authors responded these concerns specifically and added many convincing data in revised manuscript. I have no questions.

Response: Thank you for this comment!

Referee #3 (Remarks for Author):

My concerns have been addressed by the authors

Response: Thank you for this comment!

Reference

Ambady S, Malcuit C, Kashpur O, Kole D, Holmes WF, Hedblom E, Page RL & Dominko T (2010) Expression of NANOG and NANOGP8 in a variety of

undifferentiated and differentiated human cells. *Int J Dev Biol* 54: 1743–1754

Chang B, Park MJ, Choi SI, In KH, Kim CH, Lee SH (2017) NANOG as an adverse predictive marker in advanced non-small cell lung cancer treated with platinum-based chemotherapy. *Onco Targets Ther.* 19;10:4625-4633.

Chiou S, Wang M, Chou Y, Chen C, Hong C, Hsieh W, Chang H, Chen Y, Lin T, Hsu H, et al (2010) Coexpression of Oct4 and Nanog Enhances Malignancy in Lung Adenocarcinoma by Inducing Cancer Stem Cell-Like Properties and Epithelial-Mesenchymal Transdifferentiation. *Cancer Research* 70: 10433–10444

Eberle I, Pless B, Braun M, Dingermann T, Marschalek R (2010) Transcriptional properties of human NANOG1 and NANOG2 in acute leukemic cells. *Nucleic Acids Res.* 38(16):5384-95.

Koh YW, Han J, Haam S & Jung J (2019) ALDH1 expression correlates with an epithelial-like phenotype and favorable prognosis in lung adenocarcinoma: a study based on immunohistochemistry and mRNA expression data. *J Cancer Res Clin Oncol* 145: 1427–1436

Li L, Yu H, Wang X, Zeng J, Li D, Lu J, Wang C, Wang J, Wei J, Jiang M, et al (2013a) Expression of seven stem-cell-associated markers in human airway biopsy specimens obtained via fiberoptic bronchoscopy. *J Exp Clin Cancer Res* 32: 28

Li X, Yang X, Zhang G, Wu S, Deng X, Xiao S, Liu Q, Yao K & Xiao G (2013b) Nuclear β -catenin accumulation is associated with increased expression of Nanog protein and predicts poor prognosis of non-small cell lung cancer. *J Transl Med* 11: 114

Lin HW, Chiang YC, Sun NY, Chen YL, Chang CF, Tai YJ, Chen CA, Cheng WF (2019) CHI3L1 results in poor outcome of ovarian cancer by promoting properties of stem-like cells. *Endocr Relat Cancer.* 1;26(1):73-88

Maiuthed A, Chantarawong W & Chanvorachote P (2018) Lung Cancer Stem Cells and Cancer Stem Cell-targeting Natural Compounds. *Anticancer Res* 38: 3797–3809

Noh KH, Kim BW, Song K, Cho H, Lee Y, Kim JH, Chung JY, Kim JH, Hewitt SM, Seong SY, et al (2012) Nanog signaling in cancer promotes stem-like phenotype and immune evasion. *J Clin Invest* 122: 4077–4093

Palla AR, Piazzolla D, Abad M, Li H, Dominguez O, Schonhaler HB, Wagner EF & Serrano M (2014) Reprogramming activity of NANOGP8, a NANOG family member widely expressed in cancer. *Oncogene* 33: 2513–2519

Park SW, Do HJ, Choi W & Kim JH (2020) Fli-1 promotes proliferation and

upregulates NANOGP8 expression in T-lymphocyte leukemia cells. *Biochimie* 168: 1–

9

16th Nov 2020

Dear Hongbin,

Thank you for the submission of your revised manuscript to EMBO Molecular Medicine. We have now received the enclosed report from referee #1 who re-reviewed your manuscript. As you will see, this referee remains unconvinced that NANOG is expressed in human lung adenocarcinoma at a biologically meaningful level. However, this referee agrees (as we do) that your work should still be reported in EMBO Molecular Medicine, considering that NANOG is an important stem cell transcription factor, and that your results may be important to understand the differences between mouse and human lung cancer pathogenesis.

I am thus pleased to inform you that we will be able to accept your manuscript pending the following final amendments:

1) Referee #1's comments:

Please carefully address all referee #1's comments. In the discussion, please discuss along the lines indicated by this referee and discuss the value of your work for human lung pathogenesis.

2) Main manuscript text:

- Please answer/correct the changes suggested by our data editors in the main manuscript file (in track changes mode). This file will be sent to you in the next couple of days. Please use this file for any further modification.

- Please note that all corresponding authors are required to supply an ORCID ID for their name upon submission of a revised manuscript. We note that the ORCID identifier is still missing for Peng Zhang.

- Please complete the funding information in the submission system. These should match the information provided in the manuscript.

- Please remove the red text.

- Material and methods:

o Patients and tissue samples: please include a statement that informed consent was obtained from all subjects and that the experiments conformed to the principles set out in the WMA Declaration of Helsinki and the Department of Health and Human Services Belmont Report.

o Mice experiments: Please indicate the gender of the mice used for experiments.

- Please indicate in the figures or in the legends the exact $n=$ and exact $p=$ values, not a range, along with the statistical test used. Some people found that to keep the figures clear, providing a supplemental table with all exact p -values was preferable. You are welcome to do this if you want to.

- Thank you for providing a Data Availability section. However, I could not access the data. Please make sure that the data is accessible for the public before acceptance of the manuscript.

3) Figures: You uploaded a file with EV figures and legends and an EV table. The EV figure legends need to be added to the main manuscript, after the main figure legends. EV figures should be uploaded separately as individual, high-resolution figure files. The EV table should be uploaded as a separate file, in word or excel format, with its legend added directly in the file.

4) We would also encourage you to include the source data for figure panels that show essential data. Numerical data should be provided as individual .xls or .csv files (including a tab describing the

data). For blots or microscopy, uncropped images should be submitted (using a zip archive if multiple images need to be supplied for one panel). Additional information on source data and instruction on how to label the files are available at .

5) Checklist:

- section B/2: if no animals were excluded, please add a sentence to that effect.
- section B/3/b: include a statement about randomization, even if no randomization was used.
- in general, please add a few details (not only the pages of the manuscript where the information can be found).

6) The paper explained: EMBO Molecular Medicine articles are accompanied by a summary of the articles to emphasize the major findings in the paper and their medical implications for the non-specialist reader. Please provide a draft summary of your article highlighting

7) For more information: There is space at the end of each article to list relevant web links for further consultation by our readers. Could you identify some relevant ones and provide such information as well? (i.e. patient associations, relevant databases, OMIM/proteins/genes links, author's websites, etc...)

8) Every published paper now includes a 'Synopsis' to further enhance discoverability. Synopses are displayed on the journal webpage and are freely accessible to all readers. They include a short stand first (maximum of 300 characters, including space) as well as 2-5 one-sentences bullet points that summarizes the paper. Please write the bullet points to summarize the key NEW findings. They should be designed to be complementary to the abstract - i.e. not repeat the same text. We encourage inclusion of key acronyms and quantitative information (maximum of 30 words / bullet point). Please use the passive voice. Please attach these in a separate file or send them by email, we will incorporate them accordingly.

Please also suggest a striking image or visual abstract to illustrate your article as a png file 550 px-wide x 400-px high.

9) As part of the EMBO Publications transparent editorial process initiative (see our Editorial at <http://embomolmed.embopress.org/content/2/9/329>), EMBO Molecular Medicine will publish online a Review Process File (RPF) to accompany accepted manuscripts.

This file will be published in conjunction with your paper and will include the anonymous referee reports, your point-by-point response and all pertinent correspondence relating to the manuscript. Let us know whether you agree with the publication of the RPF and as here, if you want to remove or not any figures from it prior to publication.

I look forward to receiving your revised manuscript.

Yours sincerely,

Lise Roth

Lise Roth, PhD
Editor
EMBO Molecular Medicine

To submit your manuscript, please follow this link:

Link Not Available

Photos 400-800 DPI

*Additional important information regarding figures and illustrations can be found at <https://bit.ly/EMBOPressFigurePreparationGuideline>

The system will prompt you to fill in your funding and payment information. This will allow Wiley to send you a quote for the article processing charge (APC) in case of acceptance. This quote takes into account any reduction or fee waivers that you may be eligible for. Authors do not need to pay any fees before their manuscript is accepted and transferred to our publisher.

***** Reviewer's comments *****

Referee #1 (Remarks for Author):

Comments

The authors provided additional data by analyzing public RNA-seq data and performing western blotting. The data now really convince me that NANOG is not expressed in human lung adenocarcinoma at a meaningful level. Thus, NANOG may be important for mouse lung tumorigenesis in vivo, but NANOG may not be relevant to human lung cancer pathogenesis. Although the human relevance is not clear, this work should still be reported considering that NANOG is an important stem cell transcription factor and that the work highlights the difference among species (human vs mouse) in lung cancer pathogenesis. Thus, I propose the following

suggestions:

- 1) The authors should indicate in the Abstract and in a relevant place in the text that NANOG is not expressed at a meaningful level in human lung adenocarcinoma.
- 2) Include Referee Figure 12 and Referee Figure 13 in the main Figure 3 and clearly state the results in the main text that NANOG (40kDa) seen in the hESC NANOG control sample (lane) was not seen in samples (35-40kDa) from H358, A549 and H1299 lung adenocarcinoma cell lines, which suggests that endogenous and/or functional NANOG is not expressed in human lung adenocarcinoma. The authors should also clearly state the immunohistochemistry data to acknowledge the cytoplasmic expression but not the nuclear expression of NANOG in human lung adenocarcinoma.
- 3) Include Referee Figure 14 in Expanded View (Supplement?) and state the data in the main text. Also mention that RNA expression <1 TPM/RPKM is really low.
- 4) In the Discussion, the authors should discuss the differences among species in lung pathogenesis, including lung cancer (e.g., mouse specific Sca-1), cystic fibrosis and COVID-19, which will highlight the importance of the current findings by the authors.

Please carefully address all referee #1's comments. In the discussion, please discuss along the lines indicated by this referee and discuss the value of your work for human lung pathogenesis.

We have indicated that NANOG is not expressed at a meaningful level in human lung adenocarcinoma (Line 41-42, Line 217-232, Line 355-376). Also, we have included Referee Figure 12C and Referee Figure 13 in the main Figure 3 (Figure 3D and E; Line 217-229) and Referee Figure 14 in Table EV1 (Line 229-232). We have discussed the value of our work along the lines indicated by referee #1 (Line 360-381).

The authors performed the requested editorial changes.

4th Dec 2020

Dear Hongbin,

I am pleased to inform you that your manuscript is now accepted for publication in EMBO Molecular Medicine!

Please confirm that you agree with the proposed synopsis:

"This study reveals the plasticity of LKB1-deficient tumors, and identifies the Nanog-Notch axis in regulating gastric differentiation. Combinational treatment of γ -secretase inhibitor LY-411575 and phenformin effectively blocked invasive mucinous adenocarcinoma IMA formation.

- Invasive mucinous adenocarcinoma (IMA) was promoted by Nanog deficiency in the Kras^{LSL-G12D/+}; Lkb1^{flox/flox} KL mouse model.
- Concurrent loss of NANOG and LKB1 was frequent in human lung IMA.
- Mucinous differentiation was inhibited by perturbation of the Notch pathway.
- IMA was insensitive to phenformin treatment.
- IMA formation was blocked by a combinatorial treatment of LY-411575 and phenformin."

As discussed previously, we will hold publication until you notify us.

Congratulations on your interesting work!

With my best wishes,

Lise

Lise Roth, Ph.D
Editor
EMBO Molecular Medicine

Follow us on Twitter @EmboMolMed
Sign up for eTOCs at embopress.org/alertsfeeds

*** ** IMPORTANT INFORMATION ** **

SPEED OF PUBLICATION

The journal aims for rapid publication of papers, using using the advance online publication "Early

View" to expedite the process: A properly copy-edited and formatted version will be published as "Early View" after the proofs have been corrected. Please help the Editors and publisher avoid delays by providing e-mail address(es), telephone and fax numbers at which author(s) can be contacted.

Should you be planning a Press Release on your article, please get in contact with embomolmed@wiley.com as early as possible, in order to coordinate publication and release dates.

LICENSE AND PAYMENT:

All articles published in EMBO Molecular Medicine are fully open access: immediately and freely available to read, download and share.

EMBO Molecular Medicine charges an article processing charge (APC) to cover the publication costs. You, as the corresponding author for this manuscript, should have already received a quote with the article processing fee separately. Please let us know in case this quote has not been received.

Once your article is at Wiley for editorial production you will receive an email from Wiley's Author Services system, which will ask you to log in and will present you with the publication license form for completion. Within the same system the publication fee can be paid by credit card, an invoice, pro forma invoice or purchase order can be requested.

Payment of the publication charge and the signed Open Access Agreement form must be received before the article can be published online.

PROOFS

You will receive the proofs by e-mail approximately 2 weeks after all relevant files have been sent to our Production Office. Please return them within 48 hours and if there should be any problems, please contact the production office at embopressproduction@wiley.com.

Please inform us if there is likely to be any difficulty in reaching you at the above address at that time. Failure to meet our deadlines may result in a delay of publication.

All further communications concerning your paper proofs should quote reference number EMM-2020-12627-V4 and be directed to the production office at embopressproduction@wiley.com.

Thank you,

Lise Roth, Ph.D
Scientific Editor
EMBO Molecular Medicine

Corresponding Author Name: Hongbin Ji

Manuscript Number: EMM-2020-12627